# Kinematics of Deformable Blocks: Application to the Opening of the Tyrrhenian Basin and the Formation of the Apennine Chain

**Eugenio Turco [1], Chiara Macchiavelli [2], Giulia Penza [1],\* , Antonio Schettino [1] and Pietro Paolo Pierantoni [1]**

[1] Geology Division—School of Science and Technology, University of Camerino, Via Gentile III da Varano, 62032 Camerino (MC), Italy; eugenio.turco@unicam.it (E.T.); antonio.schettino@unicam.it (A.S.); pietropaolo.pierantoni@unicam.it (P.P.P.)

[2] Group of Dynamics of the Lithosphere (GDL), Geosciences Barcelona, Geo3Bcn—CSIC, Lluís Solé Sabarís s/n, 08028 Barcelona, Spain; cmacchiavelli@geo3bcn.csic.es

\* Correspondence: giulia.penza@unicam.it

**Abstract:** We describe the opening of back-arc basins and the associated formation of accretionary wedges through the application of techniques of deformable plate kinematics. These methods have proven to be suitable to describe complex tectonic processes, such as those that are observed along the Africa–Europe collision belt. In the central Mediterranean area, these processes result from the passive subduction of the lithosphere belonging to the Alpine Tethys and Ionian Ocean. In particular, we focus on the opening of the Tyrrhenian basin and the contemporary formation of the Apennine chain. We divide the area of the Apennine Chain and the Tyrrhenian basin into deformable polygons that are identified on the basis of sets of extensional structures that are coherent with unique Euler pole grids. The boundaries between these polygons coincide with large tectonic lineaments that characterize the Tyrrhenian–Apennine area. The tectonic style along these structures reflects the variability of relative velocity vectors between two adjacent blocks. The deformation of tectonic elements is accomplished, allowing different rotation velocities of lines that compose these blocks about the same stable stage poles. The angular velocities of extension are determined on the basis of the stratigraphic records of syn-rift sequences, while the rotation angles are obtained by crustal balancing.

**Keywords:** Tyrrhenian–Apennine system; non-rigid plate kinematics; rotation models

## 1. Introduction

The peri-Tyrrhenian orogenic belt, which is formed by the Apennine Chain, the Calabrian arc, and the Sicily Chain, is the most recent expression of the geodynamic process that created the western Mediterranean basin after the Europe–Africa collision (Figure 1). Large-scale extensional tectonics, coupled with orogenic processes, formed the Tyrrhenian basin, while the thrust belt–foredeep system of the Apennine chain continued migrating towards the present-day Adriatic–Ionian foreland. The Tyrrhenian margin of the Apennine chain experienced widespread extensional tectonics, characterized by formation of several marine basins, intramontane troughs, and intense magmatism. The Tyrrhenian Sea, which developed since Middle Tortonian times, is the youngest basin of the western Mediterranean region [1]. It has been extensively studied since the 1960s. In spite of the huge amount of available data, the geodynamic evolution of the Tyrrhenian basin and surrounding regions are not yet coherently described and have been subject to controversial interpretations [2–17]. In particular, the kinematic relationships between extension in the Tyrrhenian Sea, basin formation along the Tyrrhenian margin of the Apennine chain, migration of the Apennine arcs, and volcanism still remain to be determined.

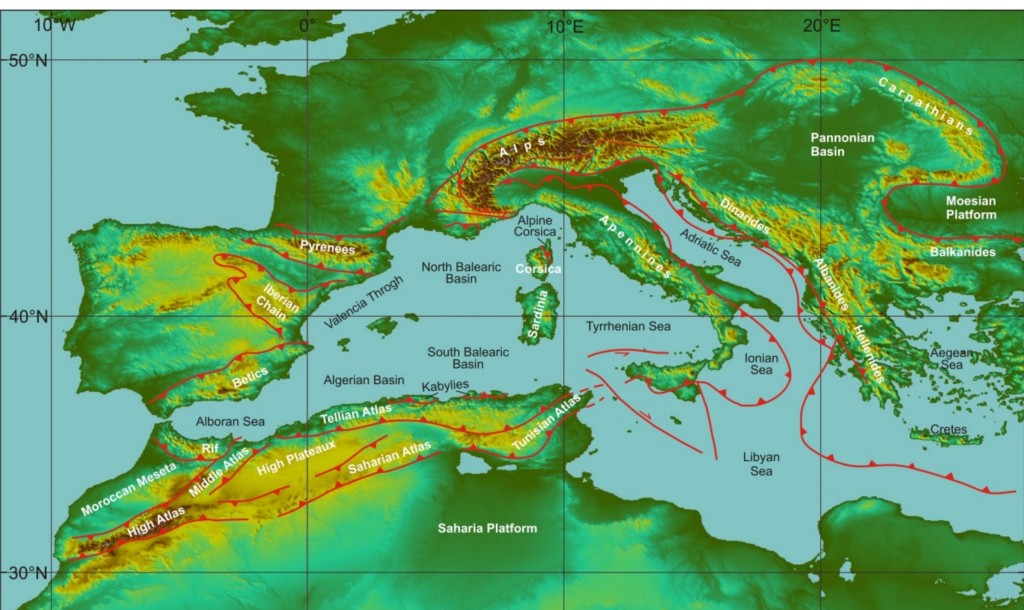

**Figure 1.** Digital terrain model (ASTER images, 1.5 s) of the western Mediterranean region with major, simplified, tectonic lineaments; modified after [18].

The main reason for the existence of controversial interpretations is due to the complexity of the geodynamic processes that generated the Tyrrhenian–Apennine system. It belongs to the Africa–Europe collision belt along which the fragmentation of the Adriatic plate started [19] since the upper Cretaceous, followed by upper Oligocene slab-retreat events. All these processes have produced an articulated Africa–Europe collision front, which includes back-arc basins and the Apennine chain. Most models proposed so far for the description of the evolution of the Tyrrhenian–Apennine system are based on stratigraphic and structural analyses of transects, not always correctly oriented along the flow lines of relative motion. Tectonic reconstructions obtained with such a 2D method often neglect important 3D kinematic constraints expressed by structures that are transversal to the chain [4,20–22]. Only a few authors have followed an approach based on the laws of plate kinematics [15,23–26].

In this work, we propose a quantitative method for describing the evolution of a system of deformable tectonic elements in the context of a back-arc extension and associated building of an accretionary wedge mountain belt. The technique is then applied to the kinematic reconstruction of the Tyrrhenian–Apennine region. Previous works [6,27,28] have described the tectonic evolution of this area in the rigid plate kinematics approximation. Here, we use the same kinematic framework but allow internal deformation of the blocks during their motion. This approach provides a better representation of the geological processes associated with the formation of back-arc basins, in particular the existence of transverse structures along the axis of the accretionary wedge. Finally, we will show that the resulting model supports the formation of at least three STEP (Subduction-Transform-Edge-Propagator) faults along the subducting slabs.

## 2. Geological Setting

### 2.1. The Apennine Chain

The Apennine–Maghreb chain is considered a Neogene thrust belt, which comprises Mesozoic to Palaeogene sedimentary rocks, derived from different basins and shelf located in paleogeographic domains of the Adria continental margin [27] (and references therein).

According to Turco et al. [18], when the Sardinian-Corsica block (SCB) started separating from the European plate, a long trench was present in the Central-Western Mediterranean region, where Liguride-Tethys lithosphere was subducting. We know that the polarity of subduction flipped from NW below the Calabrian Arc-Kabylies to SE below Alpine

Corsica (AlC) and the Western Alpine Arc (WAA). Such a structure always determines the formation of a strike-slip fault that links the two branches of the subduction zone [29]. The rotation of the Sardinian–Corsican block accompanied this process, favouring the sinistral transpressional character of the plate boundary between Adria and Sardinia-Corsica block (yellow line A, Figure 2). At the end of the Sardinia–Corsica rotation, such a lithosphere fault reached its maximum length of ~500 km (Figure 3).

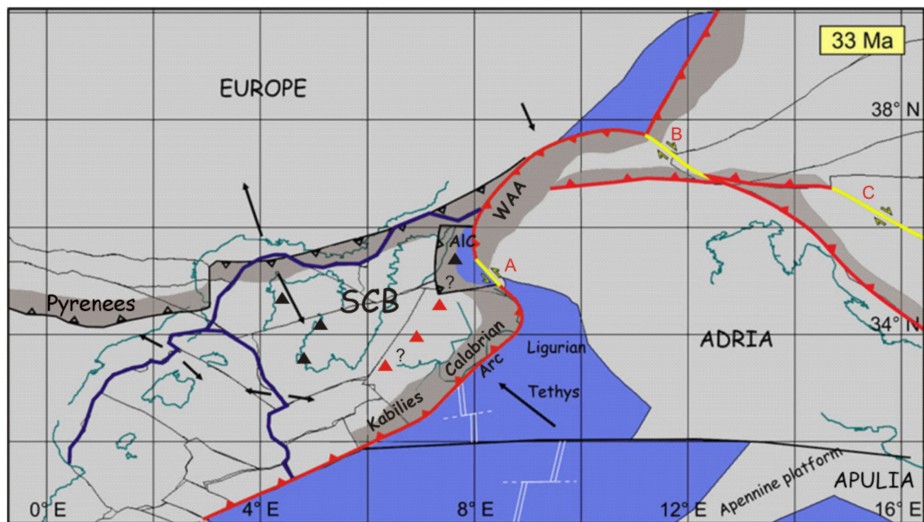

**Figure 2.** Plate reconstruction of the western Mediterranean region at 33 Ma. The distribution of the continental lithosphere is shown in (**gray**). Present-day coastlines are shown for reference. (**Black**) arrows represent direction and magnitude of relative motion. Strike–slip faults are shown in (**yellow**), labeled A, B, C. (**Red lines**) are convergent boundaries. Blue lines are divergent boundaries. (**White lines**) represent extinct spreading centers. (**Red triangles**) are volcanoes associated to the Tethys subduction; black triangles are volcanoes probably associated with the pyrenaic subduction. AlC: Alpine Corsica; SCB: Sardinian-Corsican block; WAA: Western Alpine Arc; modified after [18].

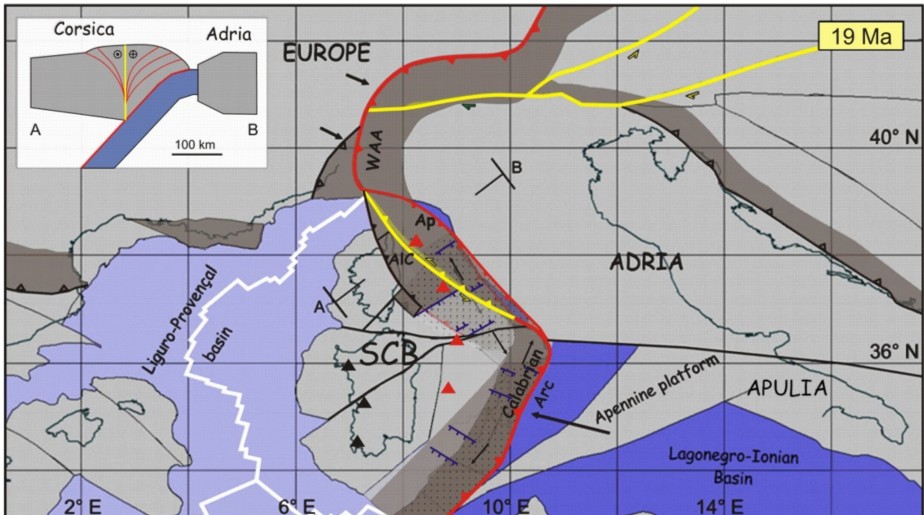

**Figure 3.** Plate reconstruction of the western Mediterranean region at 19 Ma (late Aquitanian). Dotted areas indicate wedge-top basins. Lower Miocene Chains are shown in (**dark gray**), the Africa-Adria continental lithosphere is in (**light gray**), the oceanic crust is in (**blue**). (**Red lines**) are active boundaries. (**Black lines**) are inactive boundaries. (**Yellow lines**) are strike-slip faults. (**Red triangles**) are volcanoes associated to the Tethys subduction; (**black triangles**) are volcanoes probably associated with the pyrenaic subduction. AlC: Alpine Corsica; Ap: proto-Apennine chain; SCB: Sardinian-Corsican block; WAA: Western Alpine Arc. Other symbols are the same from Figure 2; modified after [18].

At the end of the rotation, the Liguride slab was juxtaposed to the Corsica block and had dragged with it the deepest parts of the Calabrian accretion wedge, thereby a mélange of rocks belonging to the accretion wedge formed along the transpressive boundary. Complex structures, probably associated to the transpression, formed top-wedge basins, filled by sediments of the external Liguride flysch, today outcropping along the Tyrrhenian margin from Liguria to northern Calabria [30]. During the Burdigalian, the extension jumped from the western to the eastern margin of the Sardinian-Corsican block (future Tyrrhenian area) and the construction of the Apennine Chain continued further East. Therefore, the process of Tyrrhenian extension is strictly connected to the formation of the Apennine Chain. In current literature, the Apennine Chain extends from the Sestri-Voltaggio line to the Sangineto line (northern Calabria) [31–34] (Figure 4) and, on the basis of the paleogeographic domains involved in its structuring, it is subdivided in: (1) Northern Apennine, where the Ligurian allochthonous units extensively crop out [7,35–40], (2) Umbria–Marche Apennine, where sediments of the homonymous continental paleogeographic basin outcrop [7,38,41–43], (3) Lazio–Abruzzi Apennine, characterized by the presence of the homonymous Cretaceous carbonatic platform [44–49], and (4) Southern Apennine, which resulted from the deformation of the Campania–Lucania platform and the Lagonegro Basin [11,50–54]. The lateral continuity of the Apennine units is interrupted by the Calabrian arc along the Sangineto line and finds again its lateral continuity in the Maghrebide belt, which starts outcropping in Sicily, west of the Taormina line [33,55,56] (Figure 4).

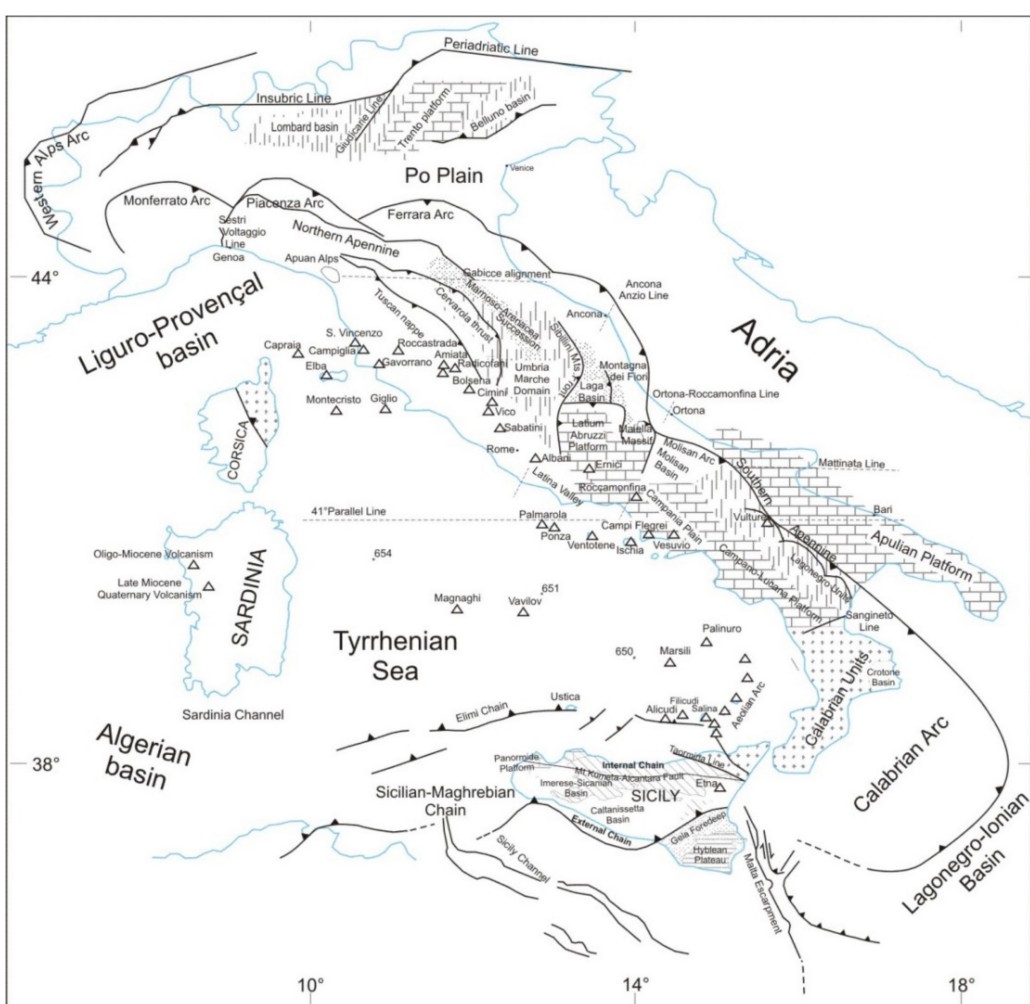

**Figure 4.** Location map of the main paleogeographic and structural units, modified after [28].

It is important to note the following key peculiarities of the Apennine chain: (a) the basement of the Adriatic margin is never involved in the structuring of the chain; (b) although

the chain is exclusively made up of carbonate rocks deriving from the Adriatic domain, the foredeep and top-wedge sediments have a composition mainly referable to continental basement rocks [57]; (c) during its structuring, while the Apennine chain grows along the Adriatic front, it is subject to extension along the Tyrrhenian side.

### 2.2. The Tyrrhenian Sea

The Tyrrhenian Sea has a triangular shape, and its northern vertex is located in the proximity of the Elba Island. Starting from the northern tip of the triangle, the Ligurian-Provençal Sea extends westwards. From the end of the 1960s and up to the entire decade of the 1980s, the Tyrrhenian basin has been the subject of many scientific cruises (DSDP, ODP, and various seismic explorations). Its formation started after the cessation of sea floor spreading in the Ligurian–Provencal basin and, according to Malinverno and Rayan and Faccenna et al. [4,10], it was due to rollback of the subducting Ionian lithosphere and migration of the Calabrian Arc towards the southeast.

An E–W lineament extending from northern Sardinia to the Campania margin, known as the 41st parallel line, is suggested to be a lithospheric left-lateral transform fault that separates the Tyrrhenian Sea in two sectors [58]. The amount and directions of extension, as well as crustal and lithospheric thicknesses, are different to the north of the line with respect to the southern region (Figure 5).

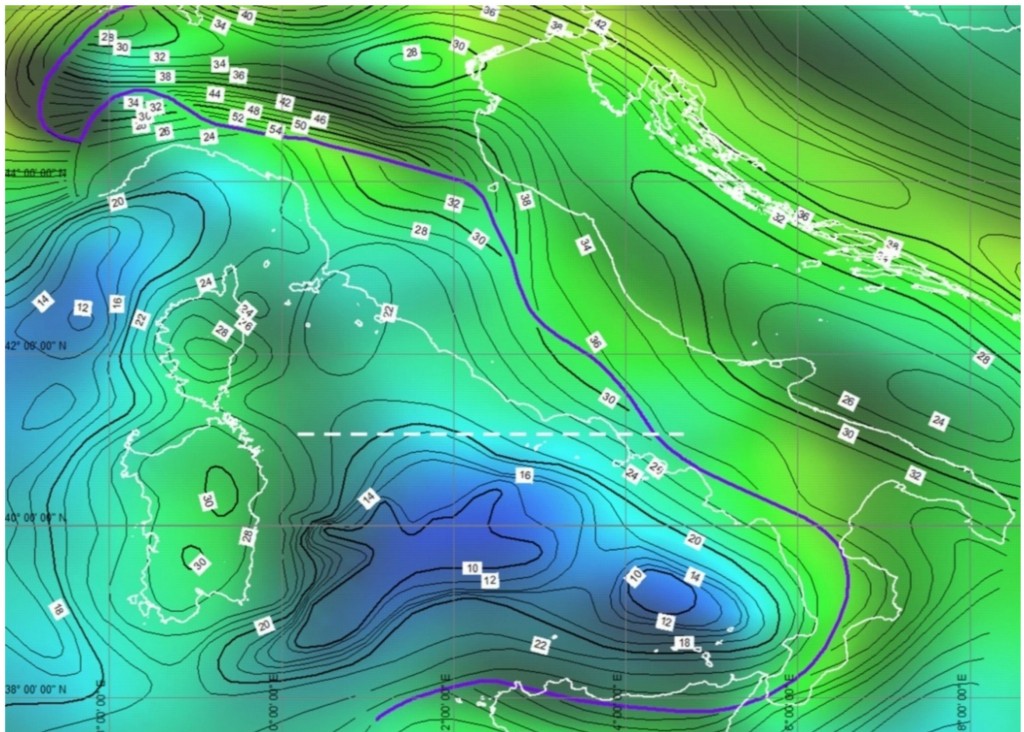

**Figure 5.** Moho map showing difference of crustal thickness to the north and south of the 41° parallel line. White dashed line is the 41°parallel line, violet line is the Moho discontinuity. [59].

In the sector north of the 41st parallel line, the continental crust is 20–25 Km thick and the lithosphere thickness is ~50–60 Km [60–62]. ODP site 654 [5] shows conglomerates covered by Tortonian, Messinian, and Plio-Pleistocene deposits. Conversely, the southern sector includes the Vavilov and Marsili basins, more than 3500 m deep, which have thin crust (25–10 Km or less) and lithosphere (30–50 Km) [60–62]. The Vavilov basin is characterized by a triangular shape, while the Marsili basin is almost squared. Both basins show magnetic lineaments comparable with the structural lineaments and the geometric shape of the basins (Figure 6).

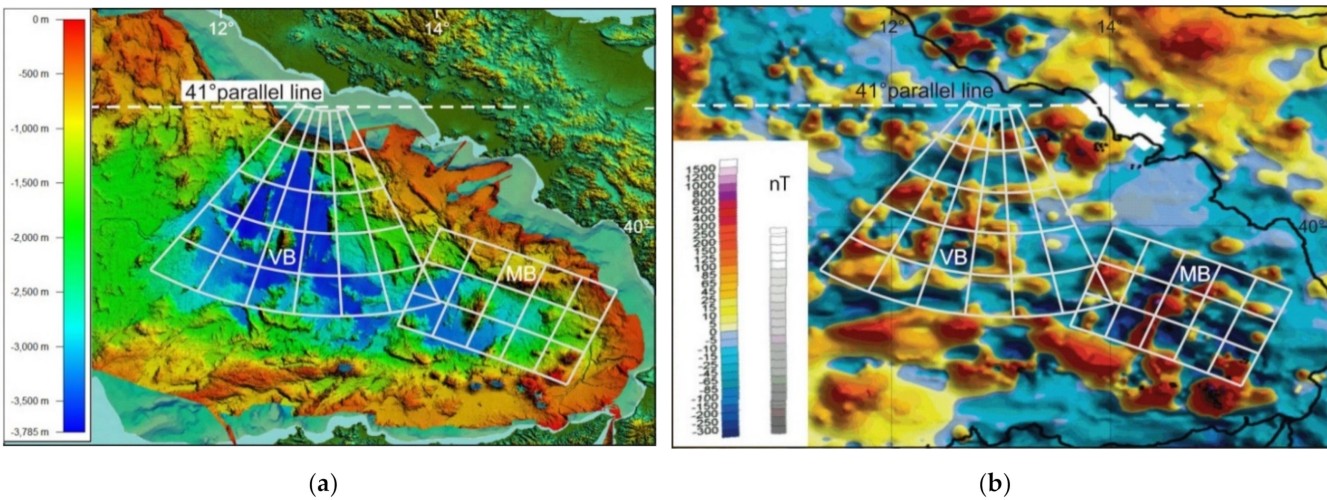

**Figure 6.** (**a**) Bathymetry [63] and (**b**) magnetic anomalies [64] for the Tyrrhenian basin. White lines show the grid of the rotation pole for: Vavilov (VB) and Marsili (MB) basins. Both basins show magnetic anomalies comparable with the structural lineaments and the geometric shape of the basins.

According to some authors [23,65–67], rifting processes in this sector started along the Sardinia margin at ~10 Ma and then migrated eastward forming the Vavilov basin first (at 5.5 Ma) and later the Marsili basin (2.0–1.8 Ma). Despite the remarkable extension, the presence of oceanic crust is likely to be restricted within these two basins [1,68,69], as testified by the ODP sites 651 and 650. The first one shows, from the top, a 388 m thick Pliocene–Pleistocene succession (bio- zone MPL6/NN18, 2 Ma) above 39-m thick succession of dolostones, lying on 29 m of highly serpentinized peridotites covered by a 134 m thick succession of basalts (lava flows and breccia). The second borehole displays, starting from the bottom, 32 m of vesicular basalts followed by dolostones and a 602-m thick succession of Plio-Pleistocene deposits (bio zone MPL6/NN18, 2.0 Ma) (Figure 7).

Despite the great commitment in the exploration of the Tyrrhenian basin, the start time of the Tyrrhenian extension process is not well constrained. According to some authors [6,70], the time of cessation of sea floor spreading in the Ligure–Provençal basin is 19 Ma. Conversely, Speranza et al. [71] proposed that the rotation of the Sardinian-Corsican block ended at 15 Ma. These ages are important because they put a constrain on the timing of roll-back for the Adriatic slab. The end of sea-floor spreading in the Ligure–Provençal basin and the Algerian basin was determined by Marani [68] on the basis of marine magnetic anomalies. The end of rotation of the Sardinian–Corsica block can be also constrained by paleomagnetic data. While Marani [68] confirmed the age obtained by the analysis of marine magnetic anomalies (19 Ma) on the basis of a compilation of quality paleopoles, Gattacceca et al. [72] proposed a younger age of 15 Ma on the basis of new paleomagnetic data and 40Ar/39Ar dating. For the beginning of the Tyrrhenian extension, several authors used the ages of the stratigraphic successions from wells and seismic correlations to suggest that it took place during the Middle Tortonian times (~12 Ma) [5,66,73]. Therefore, the time elapsed between the end of the rotation of the Sardinian–Corsican block and the beginning of the Tyrrhenian rift is not sufficiently determined so far. However, it is important to note that the elapsed time between the onset of Tyrrhenian rifting and the beginning of sedimentation in the basin depends on the speed of extension in the rift and on the original thickness of the crust.

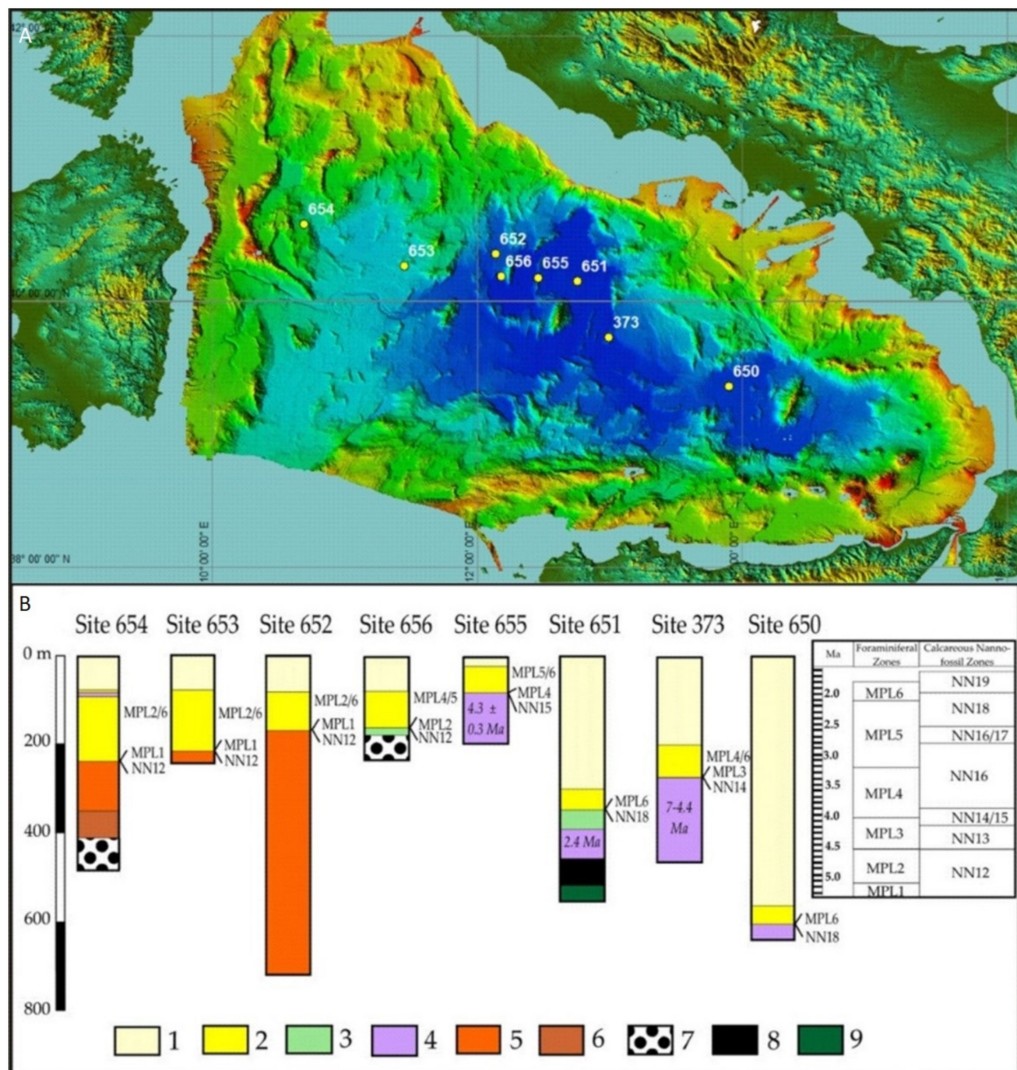

**Figure 7.** (**A**) Index map of Italy and Vavilov basin and DSDP and ODP well sites (**B**) Stratigraphic logs of the DSDP and ODP sites. (**1**) Pleistocene deposits; (**2**) Pliocene deposits; (**3**) dolostones; (**4**) basalts; (**5**) Messinian deposits; (**6**) Tortonian deposits; (**7**) conglomerates; (**8**) breccias; (**9**) serpentinized peridotites; after [69].

## 3. Methods

The fundamental tool for describing and measuring the deformation of the Earth's lithosphere, on a global scale, is the kinematics of tectonic plates. This allows to determine the path between the initial and final position of a plate with respect to another through the rotations of spherical caps about Eulerian poles [29,74] (Figure 8).

Plate kinematics has always encountered major obstacles in being used to represent deformation processes at the scale of mountain ranges, continental rifts and trascurrent boundaries, in the absence of magnetic isochrons. For this reason, palinspastic reconstructions along cross-sections at the macro-scale rely on the methods of structural geology. Retro-deformation (or crustal balancing) methods are well-known tools for the analysis of single transects, but are hardly applicable in complex areas characterized by the presence of triple junctions or polyphase systems. In these situations, the techniques of plate kinematics provide a convenient set of tools for describing the tectonic evolution and quantify the deformation of complex areas. On the other hand, structural geology can contribute to determine some fundamental parameters that must be specified to apply the plate kinematics approach, such as plate boundaries, kinematic indicators useful to determine rotation poles, and angles of rotation. It is important to specify which kind of structural data can be

used to constrain the kinematics of large tectonic blocks. For example, single slip vectors observed along the Dead Sea fault zone cannot be used for determining the rotation pole of a microplate like Sinai. In fact, due to the heterogeneity of the rocks, these data record only strains generated by local stresses. We argue instead that the structural parameters that can be useful for determining rotation poles must be observed on geological structures at the same scale of the model that is being constructed.

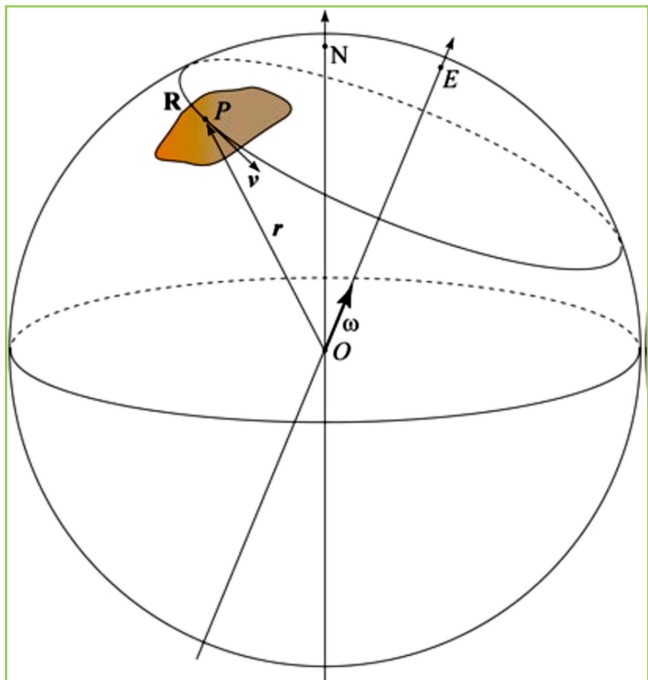

**Figure 8.** Geometry of the instantaneous motion of a tectonic plate **R**. *E* is the Euler pole, N is the North Pole. *P* is a representative point on **R**, whose instantaneous linear velocity is *v*. $\omega$ is the Euler vector of **R**; from [29].

For example, normal faults associated with an active rift such as the Tyrrhenian Sea represent a record that reflects the kinematics of extension. They exhibit sharp morpho-structural lineaments, easily observable on high-resolution DTMs or multibeam bathymetry. These features are generally devoid of "structural noise" generated by local paleo-stresses [75]. A shuttle radar topography mission (SRTM) image of the Tyrrhenian region shows lineaments that can be grouped into fan-shaped sets characterizing specific sectors of the Tyrrhenian and Apennine areas (Figure 9).

Thus, considering that these lineaments are expressions of several systems of normal faults associated with the Tyrrhenian extension, we can determine a set of Euler poles that describes the relative motion of different sectors of the Apennine chain. The classic rules of plate kinematics allow to assign these Euler poles on the basis of an estimate of the location of the convergence point of the observed lineaments [76]. All the Euler poles identified using this method describe the opening of the Tyrrhenian Sea in a European (Corsica–Sardinia) reference frame.

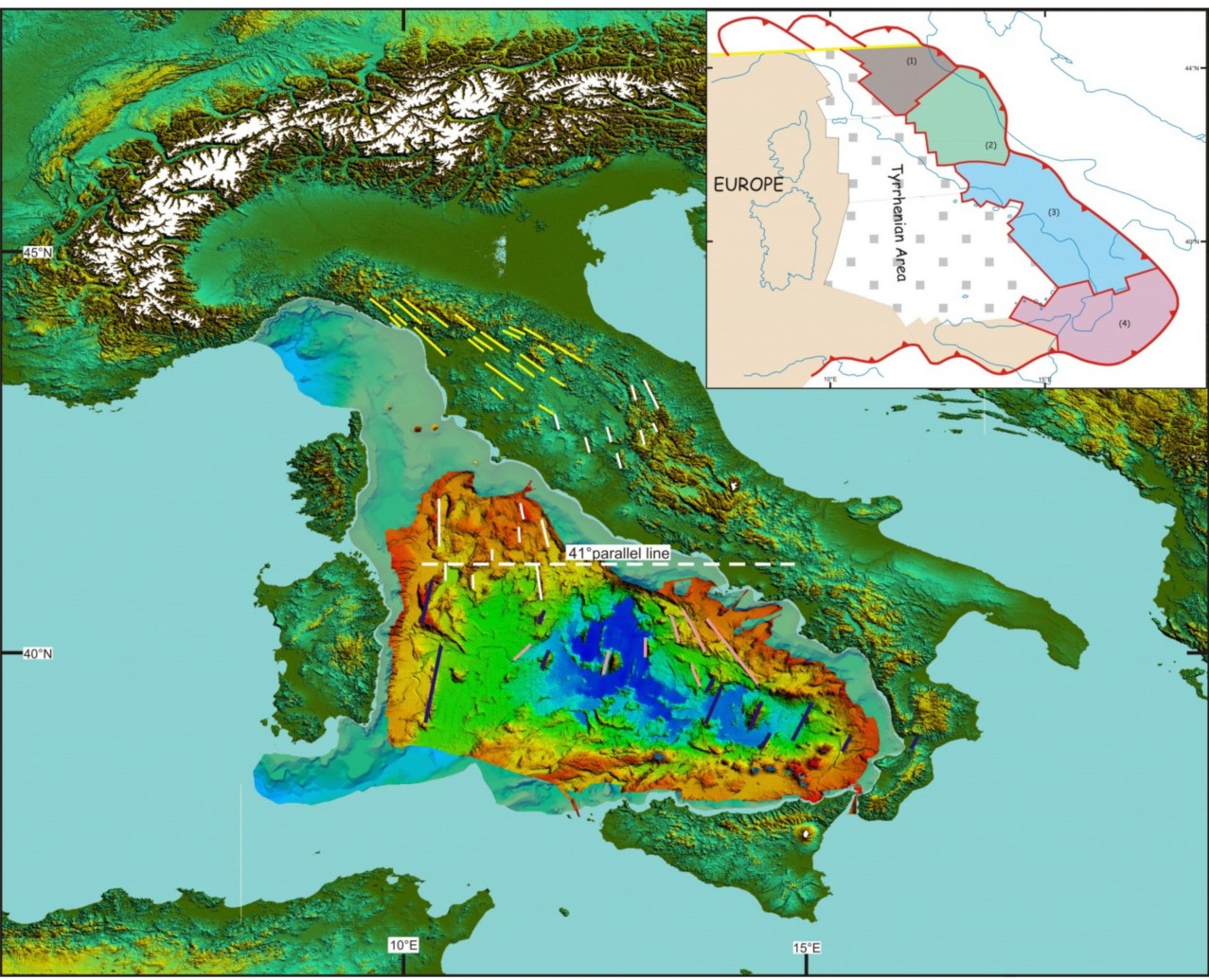

**Figure 9.** Morpho-tectonic map of the Tyrrhenian–Apennine system showing lineaments that can be grouped into fan-shaped sets characterizing specific sectors. Lineaments of the Northern Apennine sector are in (**yellow**; **white lines**) are for northern Tyrrhenian basin and Central Apennine sector; (**blue** and **pink lines**) are for Calabrian and Southern sectors respectively. The multibeam bathymetry is after [63]. The insert box shows the deformable blocks identified: (1) Northern sector in (**dark beige**), (2) Central Sector in (**green**), (3) southern sector in (**blue**); (4) Calabrian Arc Sector in (**pink**). Dotted areas are in extension; (**white dotted**) area is the extended Tyrrhenian area.

### 3.1. How to Identify Tectonic Elements in the Apennine Domain

Tectonic elements are semi-rigid crustal blocks, bounded by faults, which had an independent kinematic history in the geologic past [77]. Every single block of the Apennine area is characterized by boundaries that are divergent along the Tyrrhenian side and convergent along the Adriatic–Ionian side. They result from rotations about fixed Euler poles in the Corsica–Sardinia reference frame. The boundary between two adjacent blocks is represented by structural systems, usually transversal to the chain, that are the expression of continuously changing Euler poles in the same reference frame. The faults associated with the Tyrrhenian side boundaries represent eastward migrating fan-shaped extensional systems. On the Adriatic side, the frontal segments of the Apennine chain represent the eastern boundaries of the tectonic elements. On the basis of the lineaments identified for the Tyrrhenian area, we defined the deformable blocks shown in Figure 9, which extend along the Tyrrhenian side and grow by accretion along the Adriatic margin. Boundaries between adjacent blocks are not easily identified, because the Euler pole of relative motion is always changing and depends on the ratio of angular velocities (in the Sardinia–Corsica reference system) between pairs of tectonic elements. Boundaries with extensional kine-

matics produce systems of faults that break the continuity of the chain, forming transversal basins that are simultaneously included in the Apennine orogenic system and TTR-like and FTR-like triple junctions (Figure 10). Boundaries with strike-slip kinematics produce more complex structures and can form pull-apart basins or depressions. In some cases, for example the Ancona–Anzio line (Figure 4), the boundary can have compressive or transpressive kinematics. On the Tyrrhenian side, RRR-like triple junctions form at the intersection of rift axes (Figure 10).

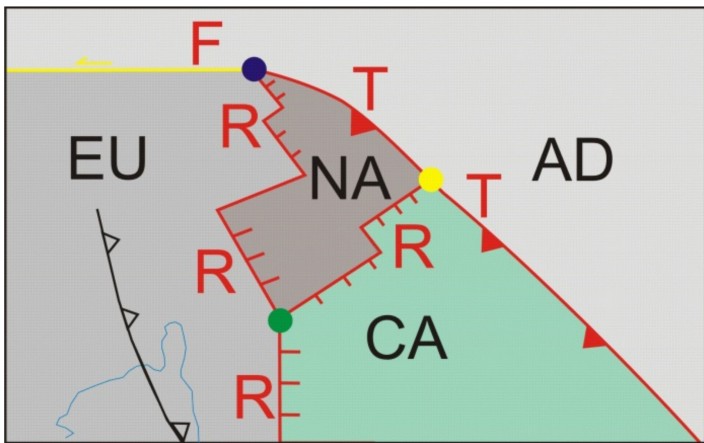

**Figure 10.** Scheme showing TTR-like (Trench-Trench-Ridge) and FTR-like (Fault-Trench-Ridge) triple junctions of the Apennine and RRR-like (Ridge-Ridge-Ridge) triple junction of the Tyrrhenian side. AD: Adria, CA: Central Apennine, EU: Europe, NA: North Apennine, T = Trench, R = Ridge, F: Fault; (**yellow dot**) is the TTR-like triple junction; (**green dot**) is the RRR-like triple junction, (**blue dot**) is the FTR triple junction.

The blocks identified in this way are highly deformable and it is not possible to identify any portion inside them that can be considered rigid. In general, the eastward migrating extensional axes overlap with some delay to the compressional structures, determining their collapse. Conversely, in global tectonics the deformed areas along plate margins represent bands with negligible surface compared to that of the entire rigid lithospheric plate. Therefore, the tectonic elements considered here behave like portions of large plate margins devoid of their rigid counterpart. However, considering the scale of the tectonic reconstructions, the internal deformation of the tectonic elements is not relevant, in our opinion, for the application of the method. The boundaries of the Apennine blocks cannot therefore be considered fixed over time. We can say that tectonic elements do not represent rigid polygons but are characterized by the rotation pole about which the lines and points contained in them rotate.

*3.2. How to Determine Rotation Poles of Apennine Blocks*

The stage poles between conjugate oceanic plates are determined building magnetic isochrons [29]. In continental tectonics, where magnetic lineaments are not available, the rotation pole between two divergent plates is generally determined qualitatively tracing central meridians of at least two parallels passing from conjugate points on the two undeformed margins [76]. Then, the Euler pole is the central point of a cloud of intersection points between meridians. Unfortunately, this method cannot be applied to the Tyrrhenian area, as it is not possible to recognize conjugate points belonging to the margins of Corsica–Sardinia and the Apennine domain, while there are no clear magnetic lineaments in the Tyrrhenian Sea. Therefore, the finite poles of rotation of the Apennine blocks can only be estimated by taking intersections of great circle arcs associated with morpho-structural lineaments of the Tyrrhenian rift. Finally, specialized software for plate kinematics (e.g., PCME-Paleo Continental Map Editor [78] or GPlates [79]) can be used to build Euler pole grids that show parallels and meridians about every single rotation

pole. Parallel circles represent flow lines of relative motion, while meridians represent the trend of extensional structures. These grids can be superimposed on the observed morpho-structures to assess the correctness of the Euler poles.

### 3.3. *How to Determine an Angle of Finite Rotation and the Start and End Times of Rotation*

The angle of rotation about a Euler pole is a function of finite strain and can be determined through crustal balancing of transects along flow lines of relative motion, granted that an estimate of the initial thickness is available. The technique is described in [29] and can be easily applied to the restoration of rifted continental margins and the reconstruction of pre-rift configurations. In the case of migrating mountain belts, the angle of rotation does not need to be determined through crustal balancing, because it can be observed directly measuring the angular distance between the present day location of a point along the thrust front and a homologous point along the estimated location of the thrust at an earlier age. Two points along the thrust front at different ages are considered homologous when the more recent point can be obtained rotating the older one about the Euler pole. In this instance, crustal balancing can be applied a posteriori to obtain the initial thickness of the chain. For example, we estimated an initial thickness of ~42 km for the central Apennine chain through the crustal balancing procedure illustrated in Figure 11. We considered a reference point in the Sardinian–Corsican block (blue point FPL in Figure 11), which was considered at the western end of the extensional area, and a homologous point along the Sibillini mountains (red point ML00 in Figure 11), representative of the eastern end the rift. The angular distance between these two points is 13°, corresponding to 180 km along the small circle arc linking the two points. The crustal balancing procedure was performed using the Moho depths of [59] (Figure 5), which have an uncertainty of 2–4 km in this area.

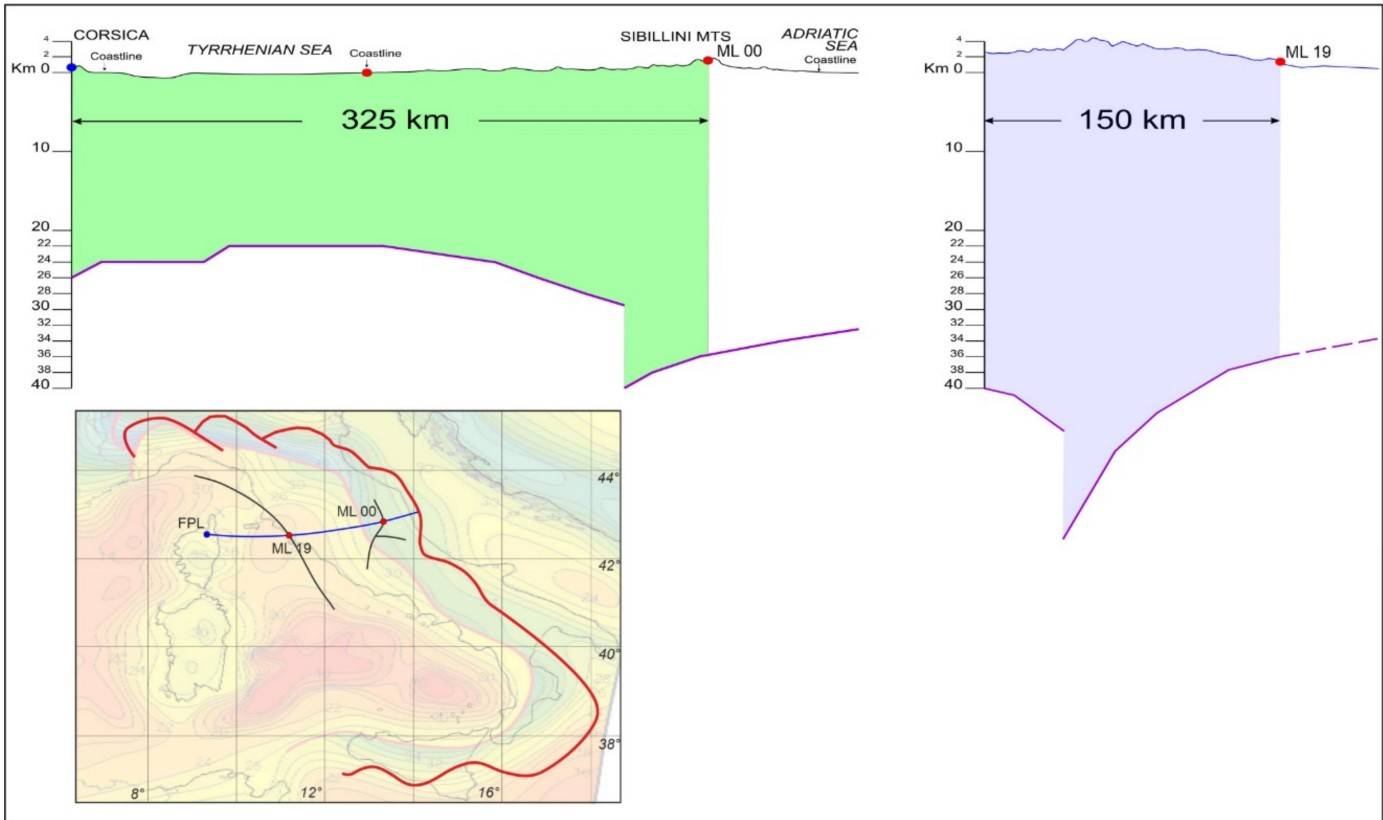

**Figure 11.** Crustal balancing technique for the Central Apennine. FPL: Fixed Pin Line (**Blue**); ML19: Mobile Line (**Red**), corresponding to the eastern extensional margin at 19 Ma; ML00: Mobile Line, corresponding to the active extensional margin.

To determine the velocity vectors, it is also necessary to know the timing of deformation. This parameter can be estimated by the analysis of stratigraphic successions that record the syn-rift tectonic activity. There is a wide array of literature for the Tyrrhenian basin with data deriving from well stratigraphy, seismic surveys, and dredging (see Figure 7) [1,67,80–82]. Furthermore, there are many other data on successions of the Apennine and foredeep basins that have recorded the time of tectonic activity of the chain [39,83].

### 3.4. Apennine Chain and Tyrrhenian Sea Sectors

We distinguished six groups of homogeneous structural systems, which characterize chain sectors. From north to south, they are: (1) Northern Apennine; (2) Umbria–Marche Apennine Arc; (3) Southern Apennine; (4) Calabrian Arc; (5) Sicilian Chain [27].

### 3.4.1. The Northern Sector

The Northern sector extends north of Elba Island and includes eastern Ligurian Sea, Northern Apennine and the area of the Tuscan rift. We consider the Northern Apennine as the northern limb of the Umbria–Marche Apennine Arc, running from the Sestri–Voltaggio line to the Gabicce alignment (Figure 4) and formed by chaotic sediments deriving from oceanic covers and boudins-like structures of Liguride ophiolites and their thick terrigenous top-wedge successions. The chain front of this sector is buried under Po river valley sediments. In the Corsica basin, syn-rift sediments are aged between Messinian and Oligocene [1,82,83]. The rift area is also characterized by the well-known Tyrrhenian-Tuscan magmatism; the age of the magmatism grows from east to west. From the structural point of view, this sector shows a fan of lineaments converging towards NW. The resulting Euler pole is located at 45.45° N, 8.40° E (Figure 12).

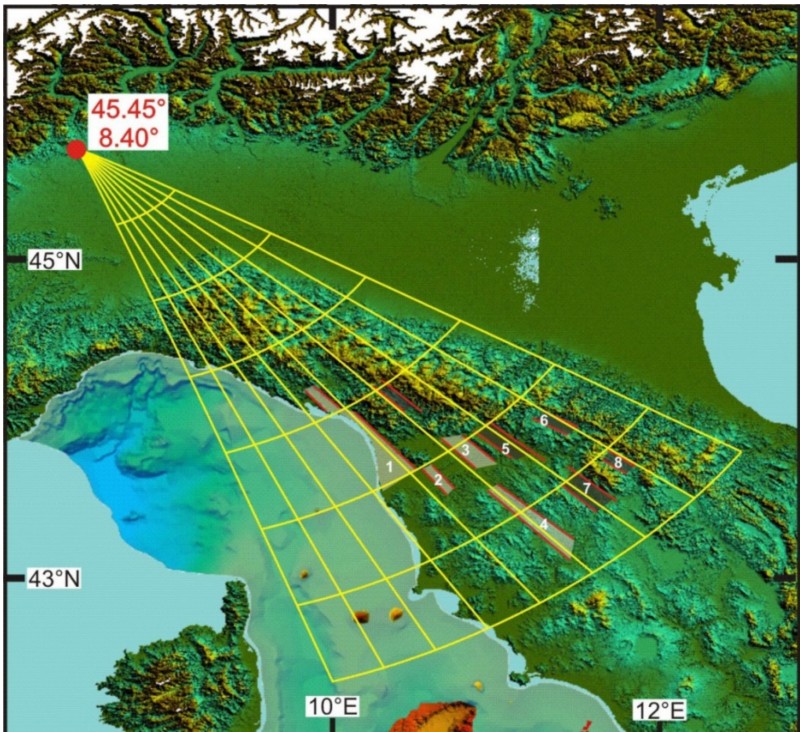

**Figure 12.** Morpho-structural lineaments of the northern Apennine and Euler pole of relative motion with respect to Sardinia-Corsica (**red dot**). The Euler pole grid shows the goodness of fit between kinematic model and geological structures. Some basins associated with the Tyrrhenian extension are also shown. Mio-Pliocene basins are in (**light gray**), Plio-Pleistocene basins are in (**dark gray**). (1) Viareggio basin, (2) Volterra basin, (3) Elsa basin, (4) Siena basin, (5) Firenze basin, (6) Mugello basin, (7) Valdarno basin, (8) Casentino basin.

### 3.4.2. Central Sector

The Central Sector extends from Elba Island to the 41° parallel and includes the northern Tyrrhenian Sea, the Umbria–Marche-Abruzzi Apennine Arc and the areas of Umbria-Lazio and Southern Tuscan rifts. The Umbria–Marche Apennine Arc is located between the Gabicce alignment (Figure 4) and the Ancona–Anzio line and is composed of deposits from the Adriatic margin (the well-known Umbria–Marche succession). The Umbria-Marche succession consists of Jurassic-Middle Miocene basinal sediments. The eastern part of the arc is the Ancona-Anzio line during the first phase of evolution and lately the Maiella front, also called "Ortona-Roccamonfina line" [34,51,84] (Figure 4). Between the two lines there is a thick Mesozoic carbonatic platform succession known as Lazio–Abruzzi platform. Tyrrhenian rift structures in this sector form lineaments converging toward north (Lat. 48.9°; Long. 10.4°; Figure 13). The Lazio margin of this sector is also affected by a Pleistocene–Oligocene volcanism.

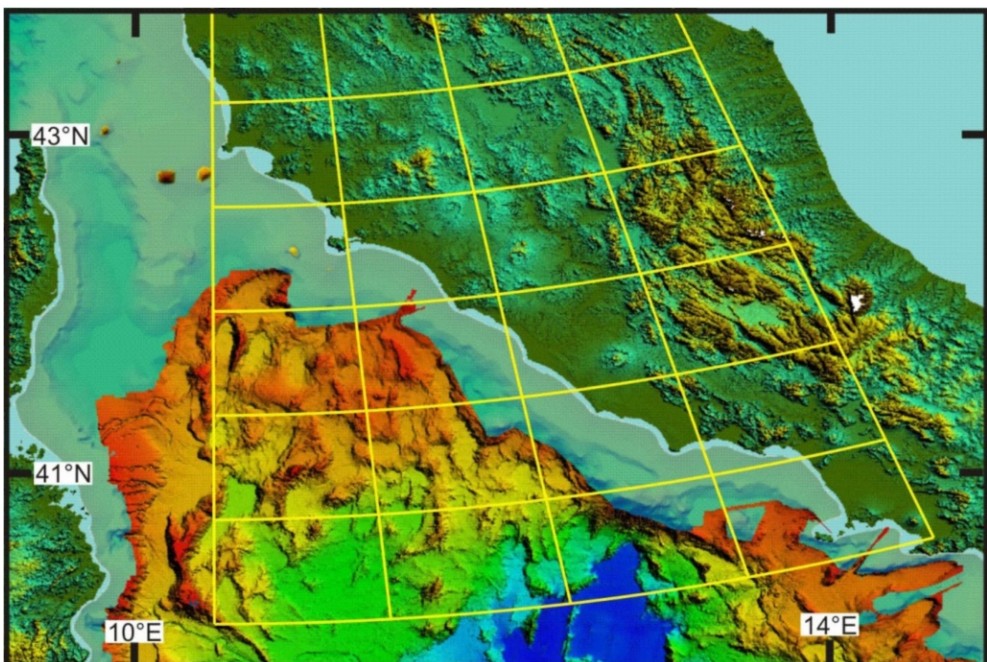

**Figure 13.** Morpho-structural lineaments of the central Apennine and Euler pole grid of relative motion with respect to Sardinia-Corsica. The Euler pole grid shows the goodness of fit between kinematic model and geological structures. The Euler pole has coordinates: 48.9°−10.4°.

The interaction between the northern and central sectors is responsible for the formation of transversal lineaments (orthogonal features) that created rectangular basins and depressions (chocolate bar), e.g., the Trasimeno Lake [85] (Figure 14).

### 3.4.3. Southern Sector

The southern sector runs from south of the 41° parallel to the Sangineto line (northern Calabria) and includes Vavilov basin and the entire Southern Apennine. The Southern Apennine is composed, from the bottom to the top, of Middle Triassic—Lower Cretaceous Lagonegro units and of carbonatic Upper Triassic—Eocene Panormide units. In some places, the Panormide units are overlapped by ophiolites and oceanic sediments of the Liguride basin, covered by lower Miocene top-wedge sediments [86,87]. On the external chain front, at the boundary with the Central sector, there is a minor arc known as "Molise arc", extended from the Maiella front to the Vulture volcano. It is made of Jurassic-Cretaceous pelagic successions and of carbonatic platforms, overlapped by Mio-Pliocene chaotic and top-wedge successions [50]. Structures of the Tyrrhenian rift of this sector form a fan of lineaments that draw the triangular shape of the Vavilov basin, whose vertex is located at 41.23° N, 13.01° E ( Figures 9 and 15).

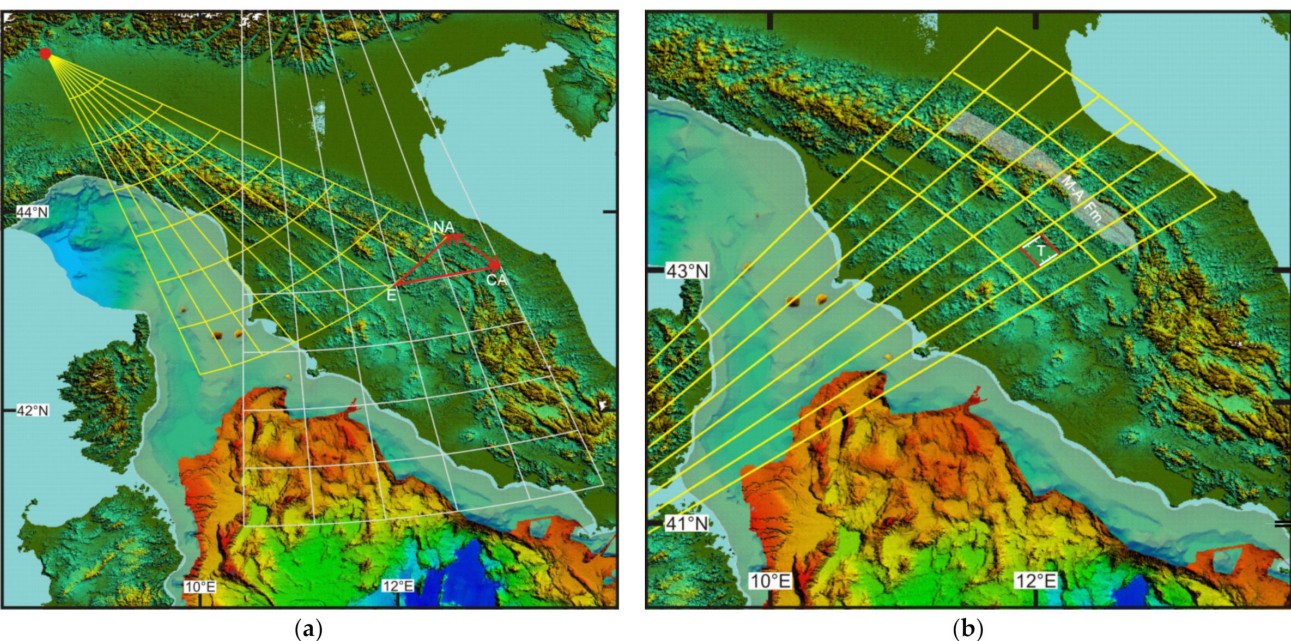

**Figure 14.** (**a**) Interaction between northern (**yellow grid**) and central (**white grid**) sectors. Red arrows are velocity vectors of the relative motion between the two sectors. E: Europe, NA: Northern Apennine, CA: Central Apennine. (**b**) Comparison between morpho-structures and the Stage pole grid valid from 5 Ma to present, between Northern Apennine and Central Apennine sectors. T: Trasimeno Lake, M-A Fm: Marnoso-Arenacea Formation. The Euler pole has coordinates: 38.95°−5.61°.

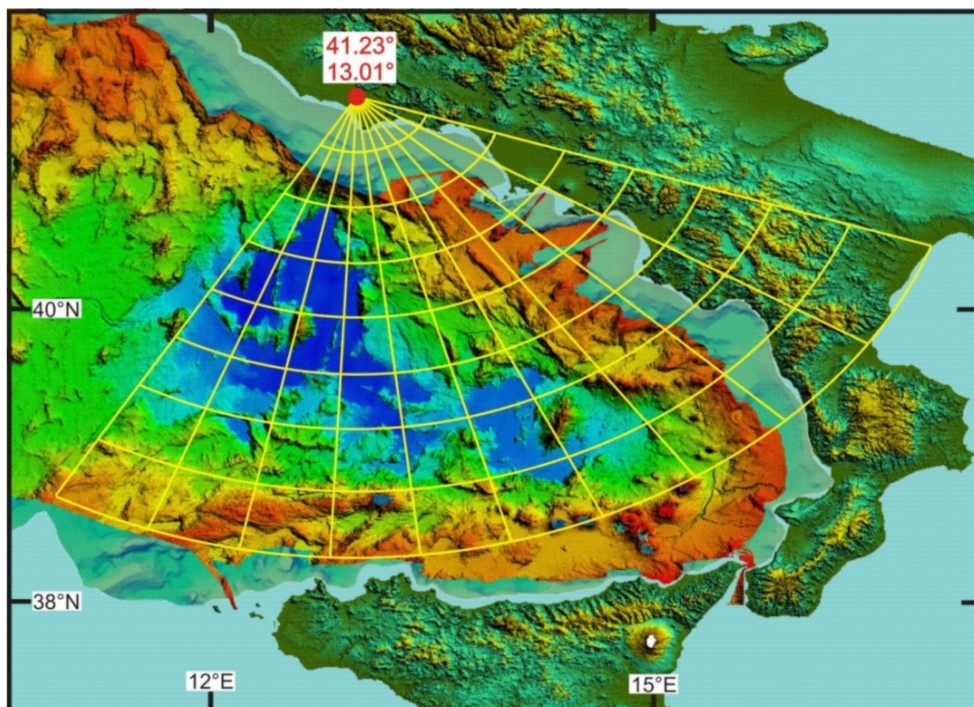

**Figure 15.** Morpho-structural lineaments of the southern sector and Euler pole of relative motion with respect to Corsica (**red dot**). The Euler pole grid shows the goodness of fit between kinematic model and geological structures.

The interaction between the southern and central sectors is responsible for the formation of transversal lineaments (ex. Gran Sasso front, Ancona–Anzio line) and basins (Laga basin) (Figure 16a). The southern sector is also characterized by complex extensional structures expressed by transverse features that cut, on the Apennine margin, the fan of the Vavilov basin. The transverse structures are the result of the interaction with the Calabrian

arc sector and form basins and depressions since the Pleistocene. Some of these contain considerable thicknesses of sediment (ex. Sant'Arcangelo basin) [88,89] (and reference therein) (Figure 16b). The southern sector also shows a strong magmatism, both on the Tyrrhenian portion and on the Campania margin. A volcano located in the Apulian foredeep of the chain (Mount Vulture) belongs to this sector.

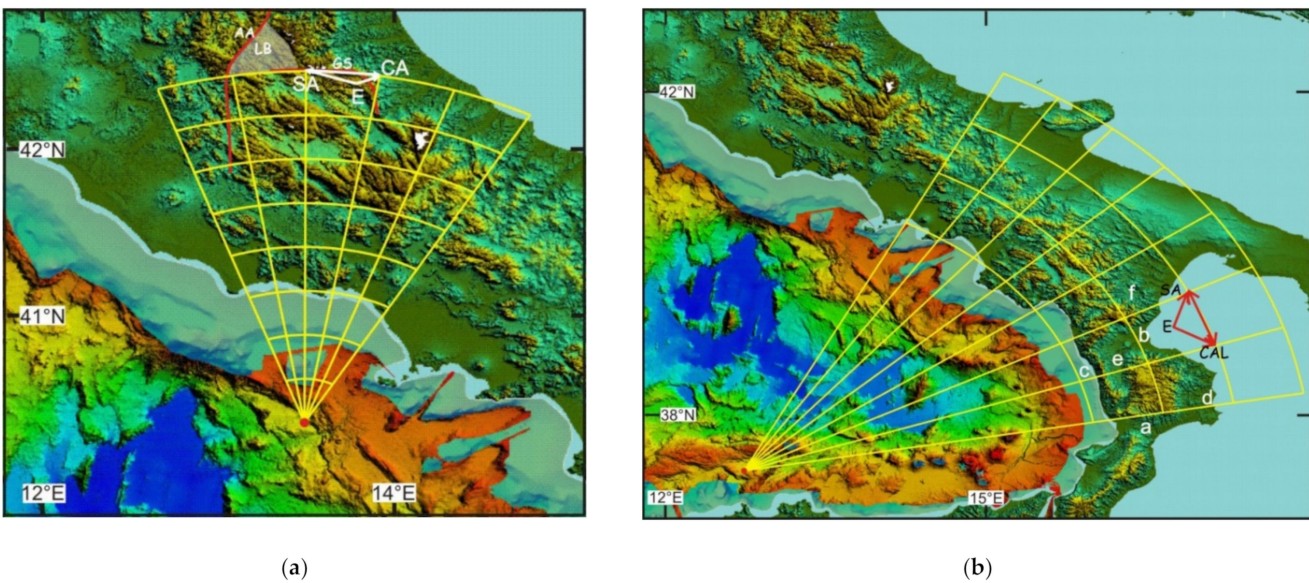

(**a**)                                                                                          (**b**)

**Figure 16.** Comparison between morpho-structures and the Stage pole grid valid from 5 Ma to present, between: (**a**) Central Apennine and southern sectors. The Euler pole has coordinates: 40.48° −13.52° (**b**) southern and Calabrian arc sectors. The Euler pole has coordinates: 38.43° −12.80°. a: Catanzaro trough, b: Sibari basin, c: Paola basin, d: Crotone basin, e: Crati valley, f: Sant' Arcangelo basin. Red and white arrows are velocity vectors of the relative motion between sectors. AA: Ancona-Anzio line, CA: Central Apennine, CAL: Calabrian Arc, E: Europe, LB: Laga Basin, GS: Gran Sasso front, SA: Southern Apennine.

### 3.4.4. Calabrian Arc Sector

The Calabrian Arc sector is bordered to the north by the Sangineto line and to the south by the Taormina line and includes the southern Tyrrhenian Sea (Marsili Basin). Structural lineaments converge towards SW. The Calabrian Arc includes the Coastal Chain, the Sila Massif, Le Serre, Aspromonte and Paleoritani mountains in Sicily. This segment of the chain is an accretion wedge formed, from bottom to top, by: Apennine carbonatic units, ophiolites (Liguride units), units consisting of low-grade metamorphic rocks with high-grade rocks on top, derived from continental crust with local flaps of Upper Trias–Upper Cretaceous sedimentary deposits. The Tyrrhenian extensional structures are represented by the lineaments that characterize the Marsili basin, the southern Calabria and the Strait of Messina. These lineaments form a mild fan with vertex towards SW (Lat. 21.85°; Long. 6.28°; Figure 17), which can be superimposed to the magnetic lineaments of the Marsili basin (Figure 6b). The Calabrian arc is also characterized by important transverse features that express extensional structures, essentially present in Northern Calabria. The most evident have generated the Catanzaro trough, the Sibari basin and also the Paola basin, the Crotonese Basin, and the Crati Valley, which are delimited by structures transversal to the arc and by high-angle N-S trending structures (Figure 16b).

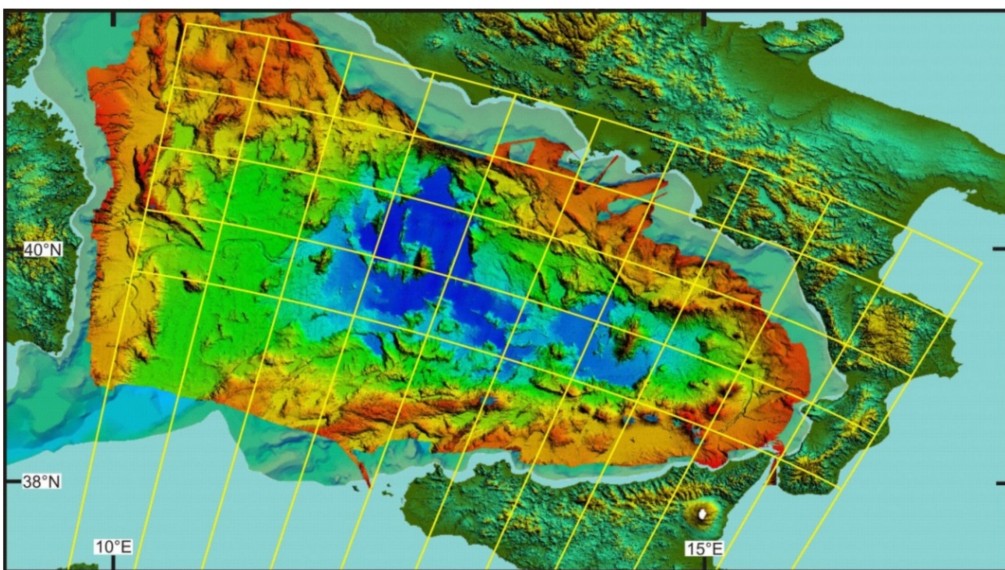

**Figure 17.** Morpho-structural lineaments of the southern Tyrrhenian and Euler pole grid of relative motion with respect to Sardinia-Corsica (**red dot**). The Euler pole grid shows the goodness of fit between kinematic model and geological structures. The Euler pole has coordinates: 21.85° 6.28°.

### 3.4.5. Sicilian Sector

The Sicilian sector is represented by the chain located west of the Taormina line. It is made up of Meso-Cenozoic basin and carbonate platform deposits, belonging to the African margin, overlapped by: sediments of the Middle Triassic-Jurassic/Cretaceous Imerese basin, sediments belonging to the Late Triassic-Eocene Panormide carbonatic platform, and Sicilidi Units covered by Oligo-Miocene sediments [90–93].

The Sicilian chain, with southern vergence is sharply divided in a northern internal sector and a southern external one by a long E-W trending lineament, known as "Monte Kumeta- Alcantara fault". This is considered a high-angle strike-slip structure with right-lateral kinematics [94]. The internal sector includes Nebrodi Mounts, Madonie and Palermo Mounts. The external sector includes Trapanese and Saccense Units, covered by sediments that filled the great depression of the Caltanissetta basin that ends south on the Gela foredeep. On the west side the foredeep is sharply interrupted in the proximity of Sciacca Mounts. To the East it passes through the Ragusa foreland and ends to the Etna Mount. The Caltanissetta basin is filled by thick Messinian and Pliocene successions and is highlighted by a strong gravimetric anomaly. This evidence suggests the presence of extensional structures that delimited the basin to the West and East [92].

The Sicilian sector, despite the Apennine sectors that show very visible and clear Tyrrhenian extensional structures, is characterized by complex structures. The Tyrrhenian offshore includes post-Messinian basins interposed between the Elimi chain and the Sicilian northern margin. The chain, E-W oriented, ends to the East with Ustica Island, where volcanic islands (Alicudi, Filicudi, Salina) mark the continuation of the lineament towards the East (Figure 4). Morpho-structures along the chain show characteristics similar to transpressive structures. Seismic profiles [92] and compressive earthquakes confirm this hypothesis. Therefore, the Tyrrhenian margin of Sicily, unlike the Apennine one, is characterized by a second phase with transpressive kinematics. To the south, the boundary with the Africa plate coincides with the Sicily channel, characterized by right transcurrent structures [95,96]. The two features delimiting Sicily converge towards west. Considering the dextral kinematic in the Sicily channel, is possible to deduce that the entire Sicily, compressed between Africa and Europa plates, is extruded towards East. The eastern side of the triangle forms the third boundary of the Sicily micro-plate. This boundary is divided in two segments: north of the Etna Mount the boundary is between Sicily and Calabrian Arc, to the south instead the boundary is between Sicily and African plate and is represented

by the Malta escarpment. Along the Malta escarpment, the Ionian slab starts to break [28], interrupting the continuity between Sicilian and Ionian lithosphere. The complexity and the numerous implications of this process deserve to be dealt in a separate paper.

### 3.4.6. Rotation Model

The tectonic reconstructions proposed here can be considered as a refinement of the regional kinematic model of the western Tethys proposed by [18,19,27,28]. They were made using an interactive computer software for plate kinematic modelling [78]. This software allows to create and maintain data sets of plate boundaries and relative plate positions, and to test the predicted plate motions through the generation of velocity fields and inferred tectonic structures.

As explained above, the classic techniques of rigid plate kinematics are not applicable in the Tyrrhenian–Apennine area. To release the block rigidity constraint, we did not use polygons to represent the different Apennine blocks. The classic polygons were replaced by linear elements that allowed to implicitly define areas with changing geometry. Therefore, each sector in the rotational model includes two separate objects that rotate about the same pole but with different angles of rotation. The western side of a block rotates with a velocity defined by the local amount of Tyrrhenian extension, while the eastern side of the arc front rotates by a higher angle that represents the accretion rate of the corresponding sector. Euler poles for the Apennine sectors are shown in Figure 18 and can be found in Table 1.

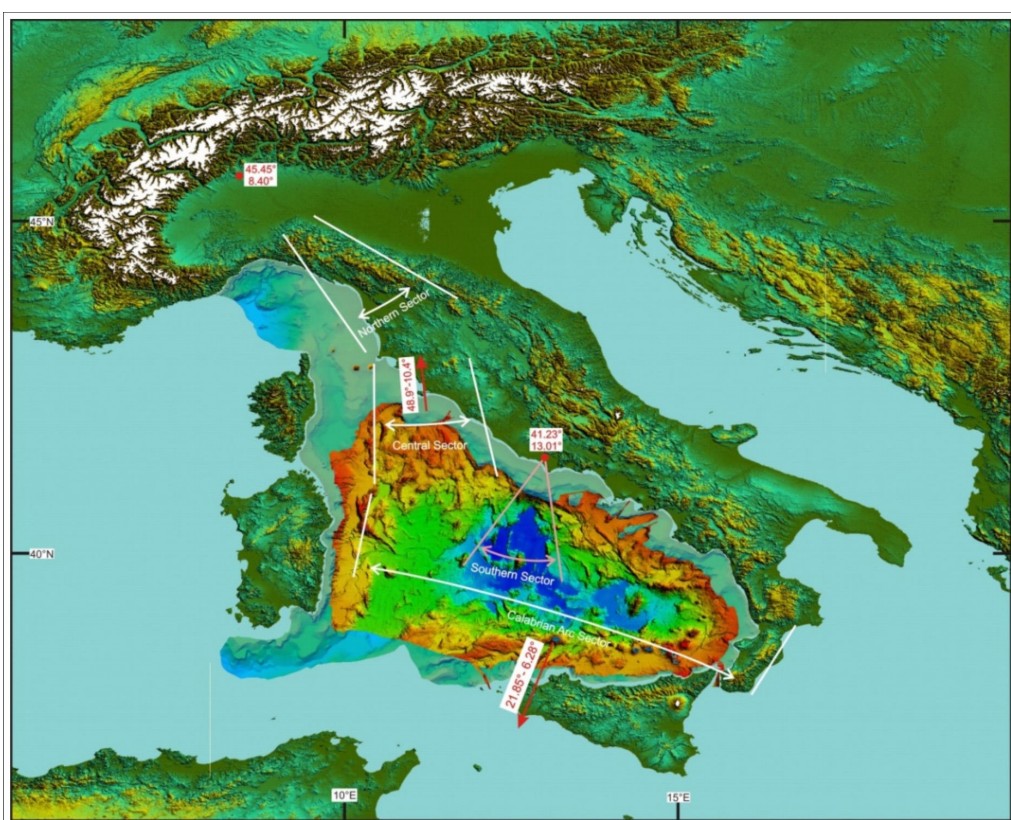

**Figure 18.** Morpho-tectonic map and multibeam bathymetry of the Tyrrhenian–Apennine system, showing extension directions in the Tyrrhenian Basin. (**White** and **pink lines**) show extension started respectively 19 Ma and 7 Ma. (**Red points**) are Euler poles. (**Red arrows**) indicate the Euler poles out of the figure, coordinates of these poles are shown. Real angle of rotation is shown for each pole. Modified after [27]. The multibeam bathymetry is after [63].

**Table 1.** Finite reconstruction parameters, from [27].

| NAME | PLATE ID | TIME [Ma] | LAT [deg] | LON [deg] | ANG [deg] | REF. ID |
|---|---|---|---|---|---|---|
| Northern Sector Rear | 384 | 19 | 45.45 | 8.40 | −20 | 301(Europe) |
| North. Sect. Front | 387 | 19 | 45.45 | 8.40 | −38 | 301 |
| Central Sector Rear | 360 | 19 | 48.90 | 10.40 | −13 | 301 |
| Cent. Sect. Front | 357 | 19 | 48.90 | 10.40 | −18 | 301 |
| Southern Sector Rear | 383 | 7 | 41.23 | 13.01 | −45.00 | 301 |
| South. Sect. Front | 390 | 7 | 41.23 | 13.01 | −54 | 301 |
| Calabrian Arc Rear | 381 | 1 | 21.85 | 6.28 | +0.46 | 301 |
| Calabrian Arc Rear | 381 | 7 | 21.85 | 6.28 | +7.0 | 301 |
| Calabrian Arc Rear | 381 | 19 | 21.85 | 6.28 | +12.00 | 301 |
| Calab. Arc Front | 388 | 1 | 21.85 | 6.28 | +0.507 | 301 |
| Calab. Arc Front | 388 | 7 | 21.85 | 6.28 | +8.75 | 301 |
| Calab. Arc Front | 388 | 19 | 21.85 | 6.28 | +15 | 301 |
| Sicilian Sector | 394 | 2 | 0.00 | 0.00 | +0.00 | 301 |
| Sicilian Sector | 394 | 2 | −21.85 | −173.72 | 1.55 | 381 |
| Sicilian Sector | 394 | 4 | −16.43 | −169.662 | 2.523 | 381 |
| Sicilian Sector | 394 | 7 | 49.408 | 33.075 | 1.398 | 381 |
| Sicilian Sector | 394 | 12 | 60.813 | 136.123 | +0.773 | 381 |

The construction of the rotation model of Table 1 (rotation model.rot in Supplementary Material) was performed as follows. Although the Euler poles associated with each tectonic stage could be determined by fitting the observed morpho-structures, we did not know a priori the corresponding angles with the exception of the rotation angle of the central Apennine with respect to Sardinia-Corsica. Therefore, we first determined the ratios between the stage angles fitting the predicted relative motion between adjacent sectors to the observed geologic structures.

Therefore, with the exception of the central sector, the rotation angles of the other tectonic elements forming the Apennine chain were determined considering the velocity ratios between adjacent blocks, which control the kinematics and the trend of the structural associations that characterize the boundaries. In practice, we chose the rotation angles in such a way that the resulting velocity ratios were compatible with the observed geological structures along the boundaries. Figures 19 and 20 summarize the timing of deformation and amount of extension on the basis of the available literature and the proposed kinematic model.

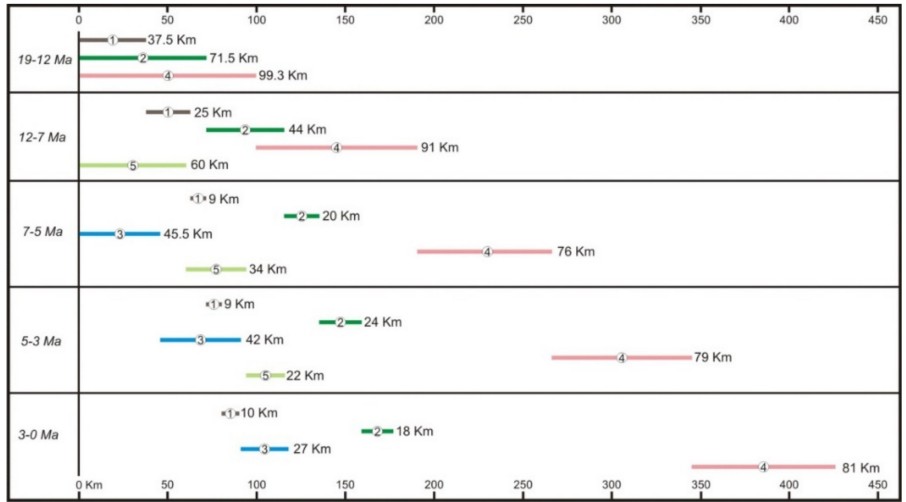

**Figure 19.** Estimated extension and timing along representative small circle arcs about the Euler poles of sectors 1–5 with respect to Sardinia-Corsica. Brown line (**1**): Northern sector, dark green line (**2**): Central sector, blue line (**3**): Southern sector, pink line (**4**): Calabrian arc sector, light green line (**5**): Sicilian sector.

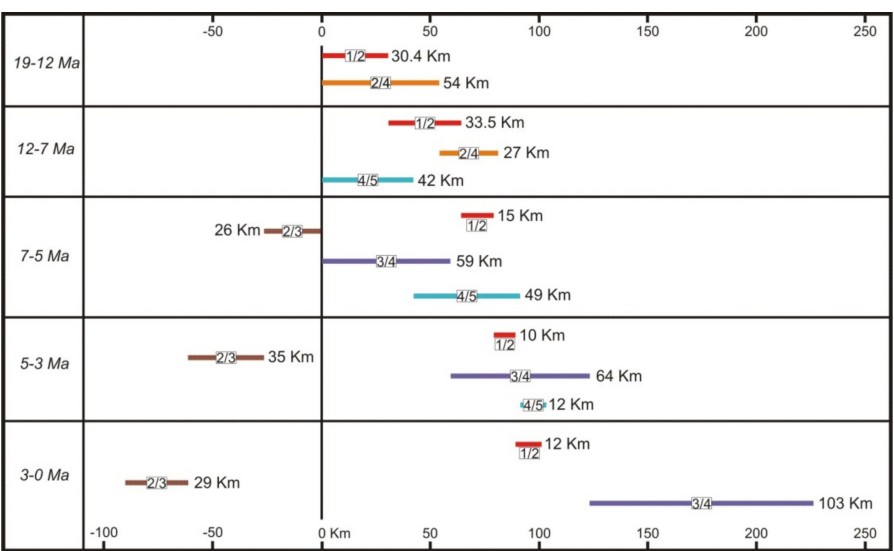

**Figure 20.** Estimated total shortening/extension and timing between adjacent sectors of Figure 19.

## 4. Results: Kinematic Evolution of the Tyrrhenian–Apennine System

### *4.1. How the Apennine Chain Evolves*

The process of evolution of the Apennine Chain is originated by two big tectonic events: the rotation of the Sardinian–Corsican block and the formation of the Tyrrhenian basin.

### 4.1.1. First Apennine Event

According to [18,70], the first phase of the Apennine evolution (33–19 Ma) generated a long left-lateral transpression along the boundary between the Adriatic Plate and the Sardinian-Corsican block. This transpression, with a major trascurrent component, drove apart the Western Alpine Arc and the Calabrian–Kabylide Arc, which aligned with opposite vergence when the rotation started (Figure 3).

At the end of this phase the transpressive structure reached its maximum length [18]. The tectonic mélange that resulted from the deformation of the accretion wedges of the two arcs, locally covered by top-wedge successions (external Liguride flysch), was the proto-Apennine Chain (Figure 3). Rocks belonging to this proto-chain outcrop in the actual Apennine chain along the Tyrrhenian margin from Liguria to northern Calabria (Falda Toscana, Apuane Alps, Argentario, Zannone island, and Cetraro and Verbicaro Units in Calabria). During this phase Adria was migrating toward NNW, while its slab was sinking into the upper mantle. As a consequence, at the end of the rotation phase the Adriatic slab was juxtaposed to the Corsica block, together with the deep portions of the Calabrian accretion wedge. The upper portion of the Calabrian wedge was still attached to the Calabrian arc.

On the southern side, the rotation phase of the Sardinian–Corsican block produced a huge stretch that was thinning the Calabrian–Kabylides Arc in its central portion. Moreover, considering the huge amount of volcanic sediments present in the internal Apennine flysch and external Ligurids successions [97], we suppose that during the rotation of the Sardinian–Corsican block a second volcanic arc was forming due to the slab of Adriatic Plate. This arc was located on the eastern side of the block attached to the first embryo of the Apennine chain.

### 4.1.2. Second Apennine Event

We describe below the Tyrrhenian phase that results from the implications of the kinematic model described above. According to this model, this phase starts at the end of the Sardinia-Corsica block rotation, with an eastward jump of the axis of extension in the future Tyrrhenian area (Figure 21 and Plate 1). The beginning time of the Tyrrhenian rifting process is not well-constrained. The oldest stratigraphic record, made up of marine

sediments in the western Tyrrhenian margin, is middle Tortonian in age [67,73] (and reference therein). Thus, the onset of rifting could be a few million years older. We assume the ending age of the Sardinia–Corsica block rotation (19 Ma) [70] as the starting age of the rifting phase, although early Burdigalian sediments can be found in the Corsica Basin, thereby a basin already existed at that time to the east of Corsica. In our interpretation, the sediments associated with the N6 biozone (18 Ma) represent the first syn-rift deposits in the Corsica basin [82], unconformably placed on older wedge-top basin sediments of the left-lateral transcurrent structure separating Corsica from Adria. However, it is important to note that the uncertainty about the starting age of Tyrrhenian extension does not affect the rotation model proposed here, because the structuring of the Apennine chain depends on the rotation speed ratios between the kinematic elements of the puzzle and not on their absolute speeds.

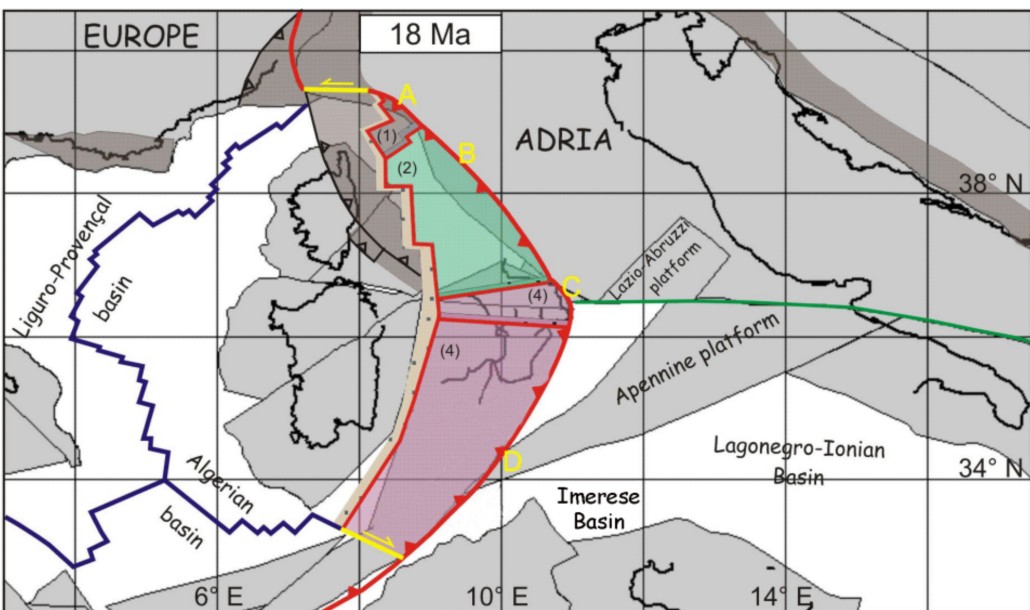

**Figure 21.** Plate reconstruction of the western Mediterranean region at 18 Ma. The distribution of the continental lithosphere is shown in (**gray**). Present-day coastlines are shown for reference. Strike–slip faults are shown in (**yellow**). (**Green line**) is an E-W directed strike-slip structure that separated Adria from Apulia from the late Cretaceous to the Eocene. (**Red**) lines are divergent boundaries, red lines with triangles are convergent boundaries. A: northern Apennine arc, B: central Apennine arc, C: northern Calabrian arc, D: southern Calabrian arc. The Northern Sector (**1**) is shown in (**dark beige**); the Central Sector (**2**) is in (**green**); the Calabrian Arc Sector (**4**) is in (**pink**). (**Light brown dotted**) area is the extended area.

The Tyrrhenian rifting presumably started behind the existing Apennine Chain, following the trend of the eastern volcanic arc (Figure 3). In the first phase of rifting, the Sardinia-Corsica margin necessarily underwent an important uplift, which inevitably caused strong erosion and further disruption of the volcanic chain already dismembered by the rifting. The Tyrrhenian extension also determined a separation between the Apennine chain and the Calabrian arc, which moved about different Euler poles with respect to the Sardinia-Corsica block. During this phase, the front of the Apennine Chain is no longer a transpressive structure, but acquires the character of an accretionary wedge. The subducting slab was made up of Adriatic lithosphere, probably teared along an E-W directed strike-slip structure that separated Adria from Apulia from the late Cretaceous to the Eocene [19] (Figure 21).

Later, the slab-retreat process produced important changes in the evolution of the Apennine chain and of the Tyrrhenian basin. We attribute these changes to new lithospheric tears that formed within the Ionian and Adriatic slabs. We can distinguish four phases

for the evolution of the Apennine–Tyrrhenian system: (1) Beginning of rifting and chain segmentation (19–12 Ma); (2) Separation of the Sicilian sector (12–7 Ma); (3) Vavilov basin formation (7–3 Ma); and (4) Marsili basin formation (3–0 Ma).

*4.2. Phases of the Evolution*

4.2.1. Phase 1. Start of Rifting (19–12 Ma)

The Tyrrhenian rift ends in the north against the left-lateral transform fault that links the Apennine chain from the Western Alps (Figure 21). To the south, the rift ends with the right-lateral strike-slip fault that separates the Calabrian arc from Kabylides (Figure 21). At the end of this first phase, the rifting process did not form important marine basins. Only on the Corsica side of the northern Tyrrhenian some marine successions can be attributed to this phase [82]. However, there are no known successions related to this phase in the southern Tyrrhenian area. As mentioned above, during this phase the Apennine chain was made of two arcs separated by a wide extension area that hosted the Sannite basin. The successions of this basin, described by Patacca & Scandone [50], form the actual Molise Arc. The model implies that the basin in this phase is located on the lithospheric flexure, generated by the tear fault between the two main slabs, which increased its subsidence (Figure 22 and Plate 1).

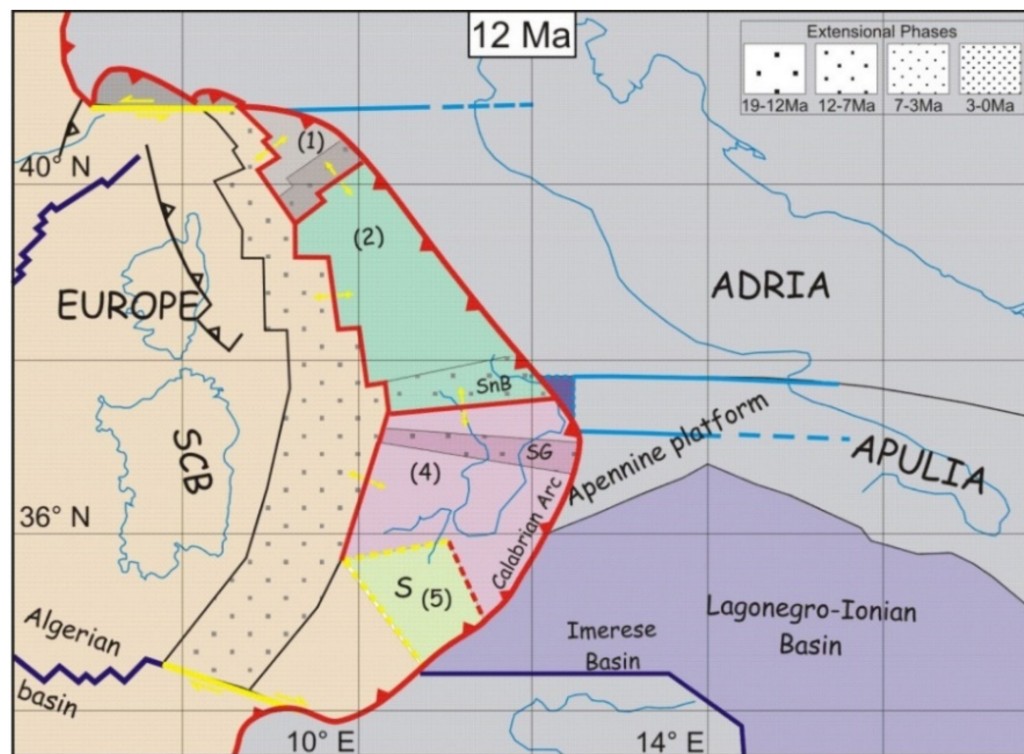

**Figure 22.** Plate reconstruction of the western Mediterranean region at 12 Ma. The Northern Sector (**1**) is shown in (**dark beige**); the Central Sector (**2**) is in (**green**); the Calabrian Arc Sector (**4**) is in (**pink**), the Sicilian Sector in light (**green**) (**5**). Dotted areas are in extension (see upper right legend). Dashed lines are incipient boundaries. (**Red lines**) are active boundaries. (**Black lines**) are inactive boundaries. (**Yellow lines**) are strike-slip faults. (**Light blue lines**) are transfer faults on the continental lithosphere and (**light blue dotted lines**) are the Jurassic COB (Continent Ocean Boundary). (**Dark blue lines**) are the Middle-Triassic COB. (**Yellow arrows**) indicate direction of extension. S: Sicily, SCB: Sardinian-Corsican block, SG: Squillace Gulf, SnB: Sannio basin.

Apennine Arc

During the first phase the arc was delimited to the north by a strike-slip fault that drove it apart from the Western Alpine arc. The fault was generated by the STEP fault of the Adriatic slab that at the same time formed an important eastward-migrating lithospheric

flexure with NW-SE trend (Figure 23), visible today in the Po river Valley. Along the flexure, an important left-lateral transpression was forming the northern Apennine sector. Geometries of the buried north Apennine structures that characterize the Po river valley are the expression of this complex process. At the beginning of the rifting, the Apennine chain was further divided in two sectors: a northern Apennine arc (A) and a central Apennine arc (B) (Figure 21), separated in transverse direction by the extensional area covered by the Marnoso–Arenacea fm [98] (Figure 16b). The central sector of the Apennine chain is associated with the Adriatic slab-retreat, accompanied by the deformation of the Umbria–Marche succession, which in this phase includes the Lazio–Abruzzi platform. Probably the transversal basins that segmented the Apennine chain were fed by sediment coming from the Liguride units, that is, units of the previous top-wedges that covered the early Apennine chain. The part of the chain affected by this process today separates the two sectors of the Apennine arc and coincides essentially with the area where the Marnoso–Arenacea formation is exposed.

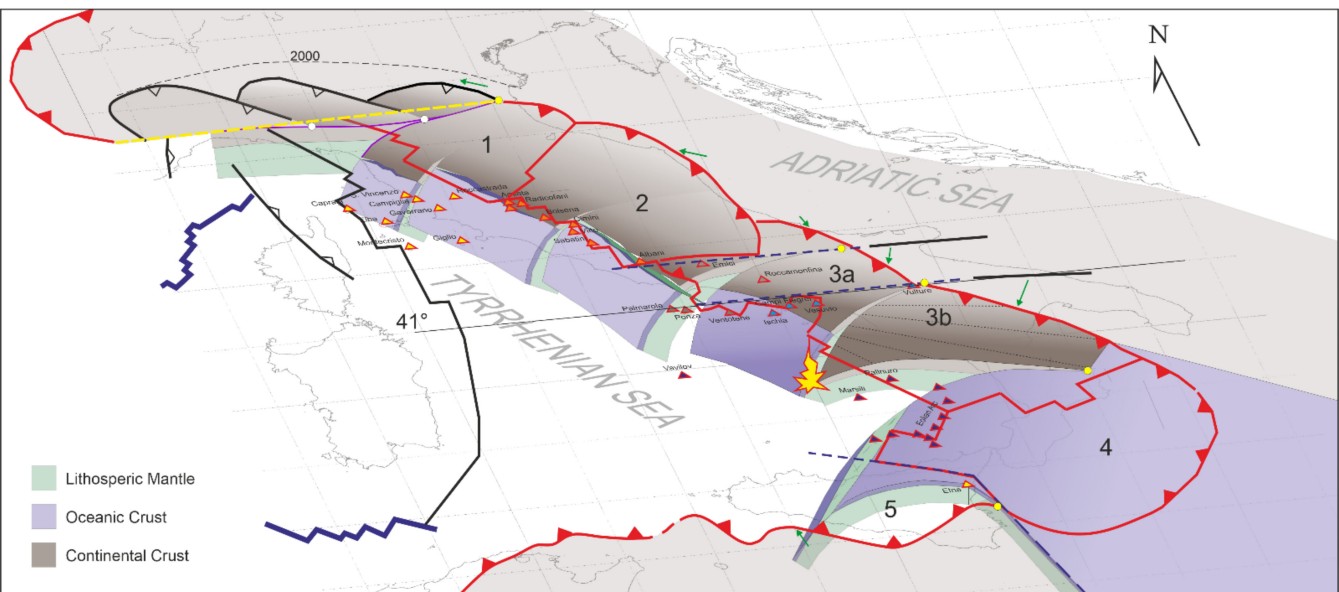

**Figure 23.** Proposed Ligurian-Ionian slab geometric reconstruction. The kinematic sectors are numbered from 1 to 5. (**Black dotted lines**) are tear faults. (**Yellow dot**) is active tip point of STEP fault (**white dot is paleo-tip point**). (**Triangles**) are volcanoes. The entire slabs length is not represented. Green arrows are the velocity vectors between Adria and the adjacent sector; modified after [28].

Calabrian Arc

The Calabrian arc is divided in two sectors: north Calabria (C) and south Calabria–Peloritani mounts (D), divided by the Catanzaro Trough (Figure 21). The transversal extension between these two sectors is recorded by late Tortonian successions in the Gulf of Squillace [73] (Figure 22). To the west, the Calabrian arc is separated by Kabylies by a large right-lateral transtensional fault that thinned the chain and delimited the Tyrrhenian rift to SW. After the subduction of the last remnant of Liguride ocean, the Calabrian wedge thrusted onto the Panormide and Imerese domains, incorporating them in the accretionary wedge. During this phase the trench of the Calabrian arc still provided an access path to the Numidian sands, which continue to fill the entire trench.

4.2.2. Phase 2. Sicilian Sector Separation (12–7 Ma)

The second phase of Tyrrhenian extension is marked by the separation of the Western Sicily chain from the Calabrian arc. The separation takes place along a right-lateral strike-slip fault that transfers the extension between Calabrian Arc and Kabylies further to the East, in the Gela foredeep. Such large transform structure that cuts the Sicilian upper plate

is known as the Monte Kumeta–Alcantara fault. It separates the Sicilian Internal Chain from the External one [90]. The western Tethys reconstructions proposed in our earlier works (e.g., [19]) imply the existence of a STEP fault along the northern margin of Sicily until the Tortonian, where Ionian lithosphere was cut away from the African margin. Here, we propose that main feature of phase 2 of Tyrrhenian extension is related to the activity of this STEP fault. The associated flexure of African lithosphere determined a southward migration of the External Sicilian chain. The external chain migrated southwards, while the Internal chain continued to migrate toward SE together with the Calabrian Arc (Figure 24 and Plate 1). The differential motion between these two sectors formed the Monte Kumeta-Alcantara fault and caused a large E-W extension in the External Chain. In the section of the chain corresponding to the Gela foredeep, the decrease in thickness associated with the lithospheric flexure of the African plate was not compensated by the thrusting of the External Chain, thus generating a strong tectonic subsidence that started to form the Caltanissetta Basin (Figure 25). The kinematic relations in the other sectors of the Chain and Tyrrhenian Rift remain unchanged in this phase. The Apennine trench passed the Carbonatic platforms and Calabrian trench reached the Ionian–Lagonegrese basin.

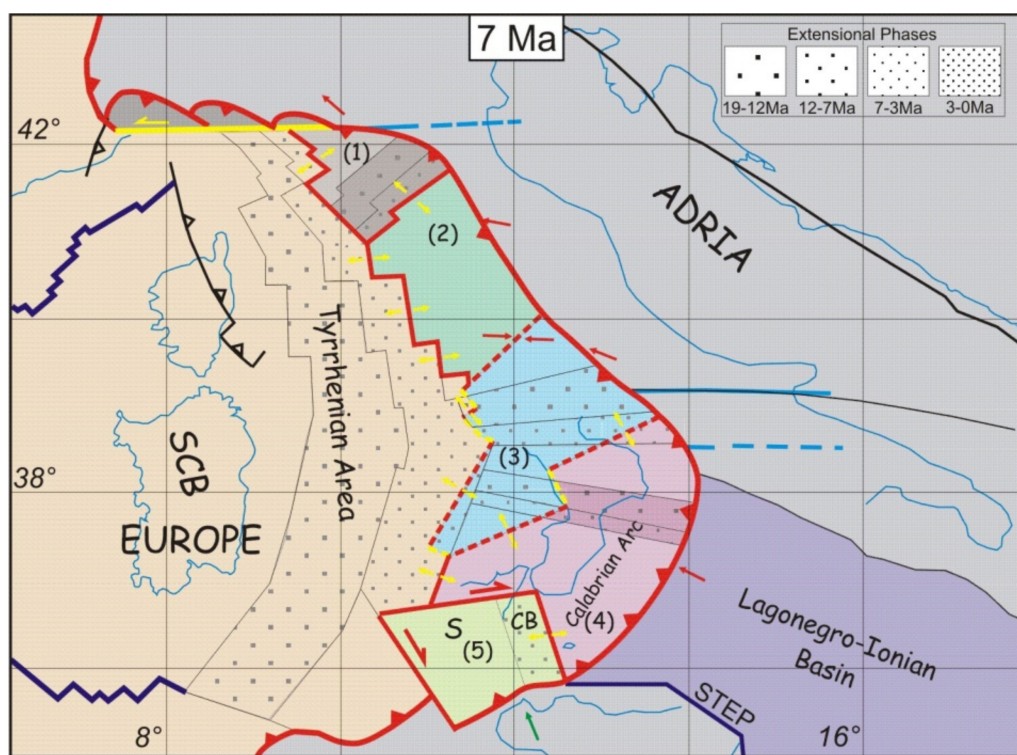

**Figure 24.** Plate reconstruction of the western Mediterranean region at 7 Ma. The Northern Sector (**1**) is shown in (**dark beige**); the Central Sector (**2**) is in (**green**); the Southern Sector (**3**) is in (**light blue**); the Calabrian Arc Sector (**4**) is in (**pink**), the Sicilian Sector in (**light green**) (**5**). (**Dotted areas**) are in extension (see upper right legend). (**Dotted lines**) are incipient boundaries. (**Red lines**) are active boundaries. (**Black lines**) are inactive boundaries. (**Yellow lines**) are strike-slip faults. (**Light blue lines**) are transfer faults on the continental lithosphere. (**Dark blue lines**) are the Middle-Triassic COB. (**Red arrows**) are the velocity vectors between Adria and the adjacent sector. CB, Caltanissetta Basin; S: Sicily, SCB: Sardinian-Corsican block.

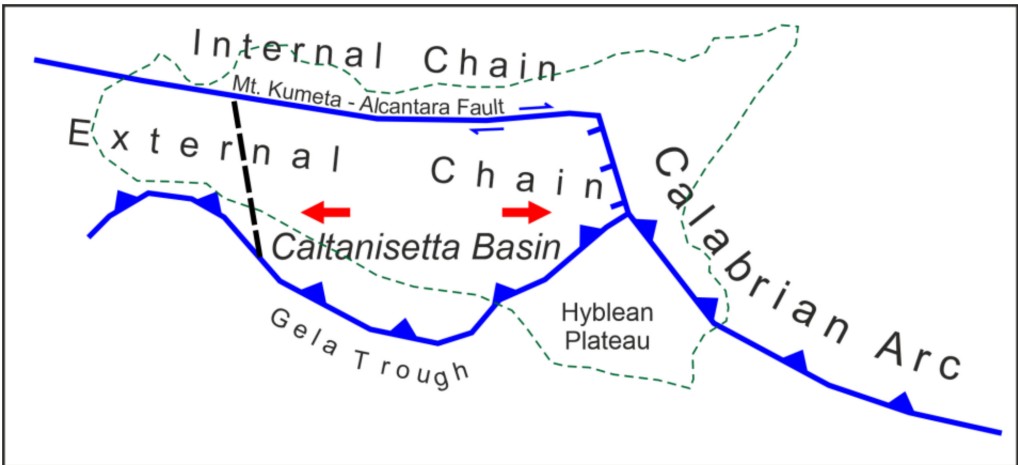

**Figure 25.** Schematic tectonic evolution of the Caltanissetta Basin. (**Red arrows**) show the direction of extension.

4.2.3. Phase 3. Vavilov Basin Formation (7–3 Ma)

　　Phase 3 of Tyrrhenian extension was characterized by the formation of the Vavilov basin, which records major changes in the Ionian slab-retreat process. At the beginning of this phase, the entire Imerese Basin and the promontory of the Panormide Carbonatic Platform were accreted to the Calabrian Arc accretionary wedge and the trench entered the narrow Ionian–Lagonegrese ocean corridor (Figure 24). The kinematic model proposed here implies the tearing of the Ionian slab along the Apulian continental margin through a STEP fault, while the eastward propagation of the 41st parallel STEP fault slowed down. On the Sicilian margin, the Ionian tear fault propagated following the continental margin and assumed a different strike. Consequently, during this phase, the units of the Lagonegro basin were also accreted to the SE migrating Calabrian wedge.

　　Figure 23 illustrates a possible present day 3D reconstruction of the Ionian slab and Apulian lithospheric flexure, based on kinematic considerations and seismic tomography [99–101]. This configuration suggests that during phase 2 the flexure of the Apulian lithosphere induced a rotation of the future southern Apennine domain determining the opening of the Vavilov Basin (Figures 26 and 27, Plate 1). The particular geometry of the three slabs shown in Figure 23 [28], whose flexure lines retreat rotating about different poles, gives the new nascent sector a striking rotation that is recorded by the simultaneous formation of the Vavilov Basin. Consequently, the Southern Apennine Sector rotated around a very close pole, located at the northern tip of the Vavilov Basin. At this stage, the Southern Apennine sector includes the Lazio–Abruzzi platform, separated from the Apennine Arc by the Ancona–Anzio Line. On the southern side, the boundary with the Calabrian Arc consists of an articulated low-angle fault that exhumes the Southern Apennines.

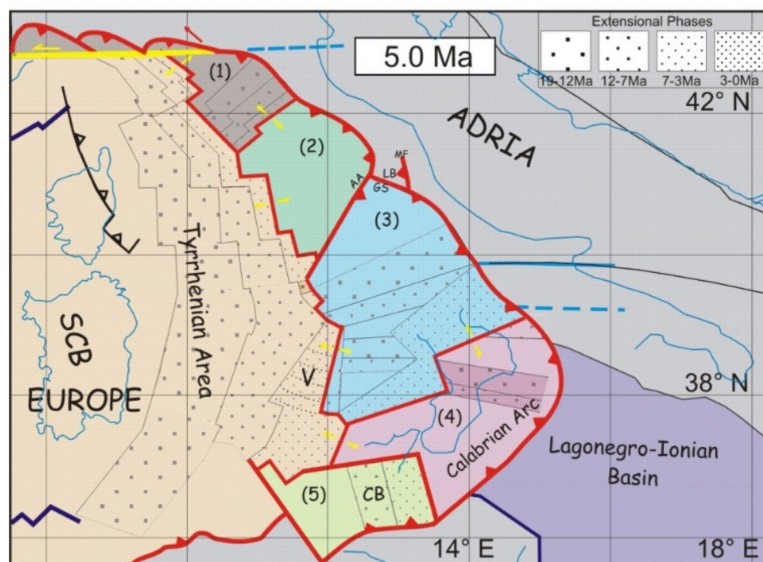

**Figure 26.** Plate reconstruction of the western Mediterranean region at 5 Ma. The Northern Sector (**1**) is shown in (**dark beige**); the Central Sector (**2**) is in (**green**); the Southern Sector (**3**) is in (**light blue**); the Calabrian Arc Sector (**4**) is in (**pink**), the Sicilian Sector in (**light green**) (**5**). (**Dotted areas**) are in extension (see upper right legend). (**Red lines**) are active boundaries. (**Black lines**) are inactive boundaries. (**Yellow lines**) are strike-slip faults. (**Light blue lines**) are transfer faults on the continental lithosphere. (**Dark blue lines**) are the Middle-Triassic COB. (**Red arrows**) are the velocity vectors between Adria and the adjacent sector. AA: Ancona-Anzio line; CB: Caltanissetta Basin; GS: Gran Sasso Front; LB: Laga Basin; MF: Montagna dei Fiori, SCB: Sardinian-Corsican block; V: Vavilov basin.

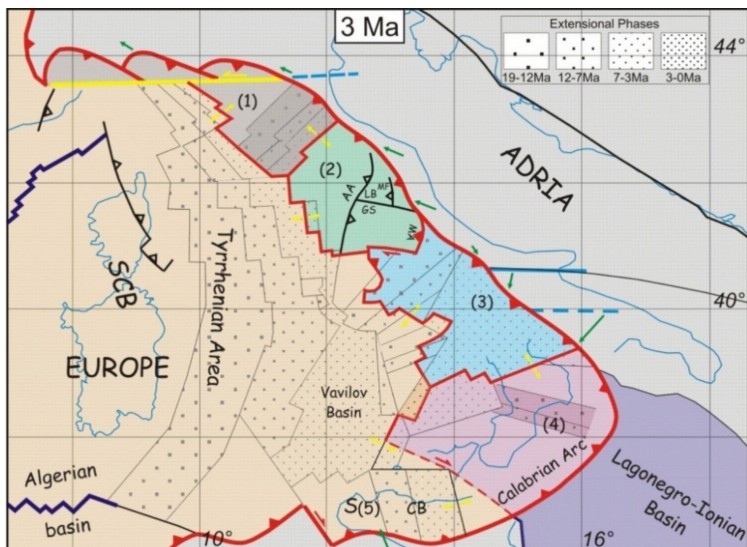

**Figure 27.** Plate reconstruction of the western Mediterranean region at 3 Ma. The Northern Sector (**1**) is shown in (**dark beige**); the Central Sector (**2**) is in (**green**); the Southern Sector (**3**) is in (**light blue**); the Calabrian Arc Sector (**4**) is in (**pink**), the Sicilian Sector in (**light green**) (**5**). (**Dashed lines**) are incipient boundaries (**Dotted areas**) are in extension (see upper right legend). (**Red lines**) are active boundaries. (**Black lines**) are inactive boundaries. (**Yellow lines**) are strike-slip faults. (**Light blue lines**) are transfer faults on the continental lithosphere. (**Dark blue lines**) are the Middle-Triassic COB. (**Green arrows**) are the velocity vectors between Adria and the adjacent sector. A-A: Ancona-Anzio line; CB: Caltanissetta Basin, GS: Gran Sasso Front; LB: Laga Basin; MA: Maiella Front; MF: Montagna dei Fiori; S: Sicily; SCB: Sardinian-Corsican block, TL: Taormina Line.

### Vavilov Basin

The Phase 3 marks the beginning of the differentiation of the Tyrrhenian basin in two areas, roughly separated by the 41st parallel lineament [58]. As mentioned above, the formation of the Vavilov Basin, located south of the lineament, arises from the rotation of the Southern Apennine Sector. The extension begins on the eastern margin and propagates with the rotation, significantly increasing the extension of the Southern Tyrrhenian Sea. The abyssal plain, more than 3500 m deep, probably made up of exhumed mantle or oceanic crust, is covered by limited thicknesses of Pliocene sediments [5]. Therefore, the abyssal area of the basin began to form in post-Messinian age. The Vavilov basin, with its triangular shape and the fan of extensional structures placed on the Apennine margin, provides the best kinematic record of the rotation of the Southern Apennine Sector. The magnetic features of the abyssal plain, on the other hand, appear less clear (Figure 6a).

### Ancona-Anzio Line

The kinematics of the Ancona-Anzio line, which represents the boundary between the Umbria-Marche Apennine and the Lazio–Abruzzi Platform, was the facies line between the Carbonate Platform and Basin during the Mesozoic. The role of this structure has been much debated [102–104]. The Southern Apennine Sector represents, according to our rotation model, a portion of the upper plate placed above three slabs. This boundary assumes an important kinematic role for the evolution of the Apennine Arc. The rotation of the two sectors around different poles generates a right-lateral transpression that cuts the Adriatic front of the chain, forming a triple junction between three transpressive trenches (Figures 22 and 23). The northern section of the trench forms the northern segment of the Ancona-Anzio line and coincides with the front of the Sibillini Mountains with eastern vergence. It represents the boundary between Adria and the Central Sector of the Apennine arc. The southern section of the line represents the boundary between the Central Sector and the Southern Apennine Sector. In this section, the transpression with western vergence is confirmed by the Antrodoco1 well [105]. The opposite vergence between the two lines gives the Ancona–Anzio line its typical flexure. The third section is outlined by the front of the Gran Sasso, which represents the boundary between the Southern Apennine Sector and Adria foreland and is characterized by right transpressive kinematics.

### Laga Basin and Montagna dei Fiori

The Ancona-Anzio line, which cut the Adriatic front of the chain above the flexure of the Adriatic slab, forms a deep depression at the triple junction, not compensated by the essentially right-lateral Gran Sasso front. The Laga Basin formed at this location, fed by the pre-Tyrrhenian Apennine flysches. At the same time, the front of the Sibillini Mountains was migrating eastwards, forming the Montagna dei Fiori with an axis orthogonal to the Gran Sasso front.

### Southern Apennine

The formation of the Southern Apennines is perhaps the most interesting result of the kinematic model proposed here. At the beginning of phase 3, the Apennine Arc and the Calabrian Arc were separated by the Sannio Basin. The tectonic units that form the actual Southern Apennines (carbonatic platform and Lagonegrese Units) represented the basal part of the accretionary wedge of the Calabrian arc. The boundary between the Southern Apennine and the Calabrian arc sectors is characterized by an articulated low-angle fault, generated by the rotations between the two sectors (Figure 16b). The activity of this fault produced a large extension that exhumed the carbonatic and Lagonegrese units from the accretionary wedge of the Calabrian Arc. At the same time, the exhumed units rotated with the Southern Apennine Sector, thereby migrating towards the Apulian flexure.

In summary, the Southern Apennine was generated by a sequence of events that took place in the following order: (1) accretion of the carbonate and Lagonegrese units in the Calabrian arc wedge; (2) exhumation from the accretionary wedge; (3) accretion of the

exhumed units to the Southern Apennine Wedge. This singular process created the space occupied by the present chain, whose front developed along a NW-SE direction.

The direction of extension that generated the exhumation of the Apennine units, according to the kinematic model, varies from NW-SE to N-S as a function of the position of the instantaneous pole of rotation between the two sectors (Figures 19 and 23). In addition, the extension is also evidenced by the formation of top-wedge basins present both in the Southern Apennines and in the Calabrian Arc (Irpino Basin, Gulf of Sibari, Crati Valley, Crotone Basin, etc.).

Taormina Line

At the end of this phase, the Internal and External Sicilian chains started moving together with respect to the Calabrian arc along the right-lateral Taormina fault. Consequently, the activity of Monte Kumeta-Alcantara fault and the extension along the External Sicilian Chain ended. The latter was previously responsible for the formation of the Gela foredeep and the Caltanissetta basin.

Northern Tyrrhenian and Apennine Arc

Along the Apennine Arc continues the rotation of the two sectors with the previous kinematics. Crustal thinning in the Northern Tyrrhenian area and between the two sectors of the Apennine Arc also continue. Tuscan magmatism is starting near the triple junction of the two rifts.

4.2.4. Phase 4. Marsili Basin Formation (3–0 Ma)

At 3 Ma extension jumped eastwards of the Vavilov basin and the southern branch of the triple junction of the Southern Tyrrhenian rifts started to form the Marsili basin in an area already thinned by transversal extension associated with the exhumation of the southern Apennines (Figure 27). At the same time, exhumation in the Southern Apennine the Lagonegro Units continued. During this phase, the Ancona-Anzio line stops its activity while, due to the propagation of the flexure of Adriatic lithosphere, the Adriatic trench jumps east of the Montagna dei Fiori and the Maiella Massif, up to the Ortona-Roccamonfina line. The Lazio–Abruzzi segment is again incorporated into the Central Sector of the Apennine Arc. The Ortona–Roccamonfina line from this moment forms the new boundary between the Apennine Arc and the Southern Apennine. The line ends North of the Roccamonfina volcano, from where it transfers a convergent movement on the southern tip of the Ancona-Anzio line through an articulated dextral transtensive structure. This E-W oriented structure runs along the Latina Valley, where the eruptive centers of the Ernici Mounts are located. The southern Apennine continues its exhumation from below the Calabrian Arc while migrating towards the Apulian flexure and generating transpessive structures in the exhumed Lagonegrese Units. At the same time, further away from the Apulian front, basins filled with marine sediments (Potenza, Santarcangelo) formed on the exhumed units still in extension. On the Calabrian Arc, the same extensional event formed the Sibari–Corigliano Basin and the Paola Basin, while affecting the existing Crati Valley and Crotonese Basin [106] (and reference therein) (Figure 16b). Further south the Catanzaro Trough was reactivated. In Sicily, the Ionian STEP fault intersected the Malta Escarpment and the tear fault began to propagate along this structure. Our model suggests that this event started widening the narrow Ionian Slab and interrupted the continuity between the Sicilian and Ionian lithosphere. All this causes dramatic changes in Sicily. During the upper Pleistocene, the triple junction of the Southern Tyrrhenian abandons the area of the Marsili and jumps near the Aeolian Arc (Figure 28 and Plate 1). The slab-retreat process from this moment is exclusively guided by the Ionian slab.

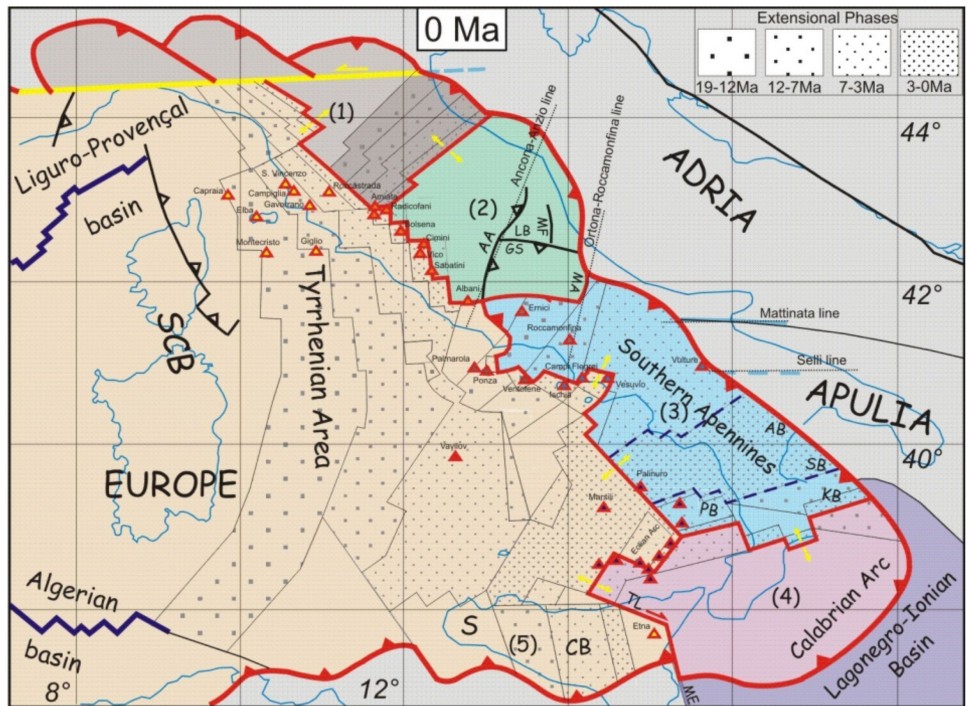

**Figure 28.** Plate reconstruction of the western Mediterranean region at 0 Ma. Triangles are volcanoes: the Tuscan province is in filled in (**yellow**), the Roman province is in orange, the Campanian province is in (**blue**), the Aeolian Arc with Marsili and Palinuro is in (**black**). The Northern Sector (**1**) is shown in (**dark beige**); the Central Sector (**2**) is in (**green**); the Southern Sector (**3**) is in (**light blue**); the Calabrian Arc Sector (**4**) is in (**pink**), the Sicilian Sector in (**light green**) (**5**). (**Dotted areas**) are in extension (see upper right legend). (**Red lines**) are active boundaries. (**Black lines**) are inactive boundaries. (**Yellow lines**) are strike-slip faults. (**Light blue lines**) transfer faults on the continental lithosphere. (**Dark blue lines**) are the Middle-Triassic COB and transfer faults on the continental lithosphere. Blue dashed lines identify the area of the Southern Apennine exhumed from 7 Ma. A-A: Ancona-Anzio line; AB: St. Arcangelo Basin; CB, Caltanissetta Basin, KB: Crotone Basin; GS: Gran Sasso Front; LB: Laga Basin; MA: Maiella Front; MF: Montagna dei Fiori; PB: Paola Basin; S: Sicily; SB:Sibari Basin; SCB: Sardinian-Corsican block, TL: Taormina Line.

## 5. Discussion

### 5.1. General Considerations

The method described in this work is based on the laws of plate kinematics and takes into account deformation of blocks through time. The proposed model describes quantitatively the complex evolution of the system of deformable tectonic elements belonging to the Tyrrhenian–Apennine region. The main tectonic consequences of this model shed new light on the geodynamic processes that generated the Tyrrhenian extension and the corresponding formation of the Apennine chain. The slab retreat process, which is generally recognized as responsible for the origin of the Tyrrhenian rift, has complex geodynamic implications, not always sufficiently considered. It arises when the speed between two converging plates is lower than the retreat speed of the slab inflection point, caused by the lithospheric sinking [107]. The process is often illustrated by a lithospheric profile that shows the kinematics of extension in the upper plate along the line of the profile. However, a correct description of the retreat of a trench should be done in 3D, considering the rotation of the trench line about the Euler pole of the upper plate. Besides the kinematic aspects, the slab-retreat process is associated with important geodynamic implications, such as the formation of STEP faults and the elastic rebound that follows slab detachment, and requires a description that takes into account of all the boundary conditions. For example, STEP faults determine a downward flexure that partly affects the unsubducted plate, whose margin is dragged to some extent in the asthenosphere. Such a flexure propagates

together with the rip between the slab and the continental margin. In the crust, the effects of this process are strictly related to the extent of the flexure, which in turn depends on the rigidity and strength of the lithosphere. In general, the marginal flexure of the continental lithosphere occurs along a hinge line that is oblique with respect to the STEP fault strike. The subsequent rebound determines the formation of crustal structures that are pairwise oblique with respect to the direction of slab retreat. In the Apennine area this effect is quite evident. In addition to this, the lithospheric flexure along a STEP fault produces important transpressive structures at crustal scale that simulate normal mountain chains. On the basis of these considerations, it is possible to describe the evolution of some sectors of the Apennine and Sicilian chains.

### 5.2. STEP Fault Evolution Within the Northern Apennine

In previous models, the northern and central Apennine are part of a unique arc that extends from the Sestri Voltaggio line to the Ortona–Roccamonfina line [108]. These models are based on the hypothesis that a continuous Adriatic slab exists. Our kinematic model suggests instead that the northern Apennine segment represents a transform structure of the upper plate with left-lateral motion, which links the western Alpine arc with the Apennine chain with opposite vergence. The presence of a mountain chain along this fault is a consequence of the geodynamic complexity of the processes involved in STEP faults. The flexure of the continental margin of Adria in the Padania Valley originated a submerged chain that was later exhumed by elastic rebound. A series of arcs formed, with a symmetry axis aligned with the hinge line of the flexure, thereby oblique with respect to the strike of the STEP fault. These arcs propagated eastwards and were eventually exhumed (the Monferrato arc) following the tear migration (Figure 4 and [109]).

### 5.3. STEP Fault Evolution Within the Sicilian Chain

Our model proposes that, differently from the northern Apennine, the western Sicilian chain developed along three distinct STEP faults. The oldest (and most important) one had E-W trend (in present day coordinates) and was placed along the northern continental Hyblean paleomargin. The next, very short STEP coincided with the modern Taormina line. Finally, the modern STEP fault associated with the rip of the Ionian oceanic lithosphere from the African margin coincides with the Malta Escarpment. The eastward propagation of the tearing determined flexure along the Iblei and SSE migration of the external chain while the internal chain moved together with the Calabrian Arc (Figure 4). This process separated the two chains along the Monte Kumeta fault, which is characterized by right-lateral kinematics. The eastern termination of this fault is represented by the northeastern tip of the Caltanissetta extensional system. During the early Pliocene slab tearing was transferred to the Taormina line, thereby starting from this time tectonic activity along the external and internal chains was essentially driven by the convergence between Africa and Eurasia. Finally, during the Pleistocene, slab tearing started along the Malta Escarpment, determining an eastern break in the continuity between the African continental margin and the Ionian lithosphere. We hypothesize that this event originated an eastward escape of Sicily with respect to stable Africa. Such a hypothesis is supported by the dextral kinematics along the Sicily Channel and by the submerged Elimi Chain [110,111]. A successive paper will address this point.

### 5.4. STEP Fault Evolution Within the Southern Apennine

The southern Apennine is aligned with an important STEP fault that separates the continental margin of Apulia from the Ionian slab. Initially, the southwestern margin of Apulia bended downwards following the subduction of the Ionian oceanic lithosphere, while its northern margin separated by Adria along the 41° parallel STEP fault. Starting from the late Tortonian formation of the Vavilov basin, a tear formed that separated Apulia from the Ionian oceanic slab. As soon as Apulia was not anymore pulled downwards by the Ionian slab, the flexure line started rotating counterclockwise and assuming a strike slightly

oblique with respect to the new southeastward propagating STEP fault. The contemporary elastic rebound of the more western parts enhanced this rotation of the flexure line and produced at the same time the rotation of the southern Apennine and the formation of the Vavilov basin. The difference in the Euler poles of rotation of the southern Apennine and the Calabrian arc with respect to Europe determined strong extension between the two sectors and exhumation of most of the southern Apennine. The latter was previously part of the Calabrian arc accretionary wedge.

## 6. Conclusions

The tectonic reconstruction of the Tyrrhenian basin and the Apennine chain offered us the opportunity to propose a method that integrates plate kinematics tools with the methods of structural geology. The technique has proved to be adequate for performing quantitative deformation analyses at an intermediate scale such as that of the Mediterranean area.

Structural geology has made significant contributions for the application of plate kinematics methods to continental tectonics. Among these, we remind the identification of the components of the Apennine puzzle (micro-plates), the determination of the poles and angles of rotation, and the definition of the plate boundaries. With these fundamental parameters, we have formulated the hypothesis of a rotation model of the puzzle of micro-plates, built with the aid of the PCME software tool. The model provided a framework of kinematic constraints that allowed to hypothesize innovative solutions to the problems inherent the tectonic reconstruction of the Apennine chain. The kinematic framework is also useful for future research that can confirm, improve, or modify the model we propose.

The model can be viewed through an animation (Movie 1 in Supplementary Material) composed by a temporal sequence of images, generated by the software. The characteristic of this approach, well known in the global tectonics studies, is to provide and visualize, for each point along plate boundaries, the relative velocity vectors between the micro-plates. Finally, this important feature allowed us to predict the tectonic and structural implications along the plate boundaries. The expected implications of our rotation model have found, in our opinion, a very satisfactory confirmation for the tectonic structure we observe for the Apennine chain and the Tyrrhenian basin. We therefore believe that the initial hypotheses made on the recognition of micro-plates, on the determination of the rotation poles, on the activity times, and on the angles of rotation have been verified. The model also provides a precise balance of the areas, which is essential for paleogeographic reconstructions.

Due to these important characteristics, we believe that the rotation models constructed with this methodology, thanks to their ability to predict implications of tectonic and structural processes in a very broad framework, represent powerful tools suitable for responding to interpretative disputes for complex areas still debated. We are however fully aware that the transition from the rotation model to its tectonic implications requires a certain aptitude to visualize the third dimension, not represented in the rotation models.

For the Apennine and Tyrrhenian areas, the rotation model has responded well to the historical issues. Among these we recall the most significant. Starting from the North, they are: the transversal structures that interrupt the Umbria–Marche Arc and separate it from the Northern Apennine, while extending the Tyrrhenian, Tuscan–Lazio and Umbria areas in a direction transverse to the Apennine chain; the Ancona-Anzio transpressive lineament which, in the triple junction with the Sibillini and Gran Sasso front, form the Laga basin; the exhumation of the tectonic units of the Southern Apennine, which implies a great extension that pervades the entire chain; the formation of important basins, located near the foredeep of the southern chain (Santarcangelo, Sibari, Crotonese and Caltanissetta), the Paola basin, and Crati Valley located in an innermost position and the Catanzaro trough that crosses the entire Calabrian arc. All these basins exhibit extension along the direction of the chain that is not appreciable in the tectonic reconstructions of transects.

**Supplementary Materials:** The following are available online at https://www.mdpi.com/article/10.3390/geosciences11040177/s1, Plate 1: Kinematic Evolution of the Tyrrhenian–Apennine system; rotation model.rot; Movie 1: Kinematic Evolution of the Tyrrhenian–Apennine system.

**Author Contributions:** Conceptualization, methodology and writing—original draft preparation, E.T.; software, investigation and data curation, C.M., A.S., G.P.; funding acquisition, supervision and writing—review and editing, P.P.P. All authors have read and agreed to the published version of the manuscript.

**Funding:** This research was funded by the Italian Ministry of University and Scientific Research, PRIN Turco, prot. 2008YWPCWB and by the Università degli studi di Camerino (Far-Pierantoni), Grant number STI000030.

**Institutional Review Board Statement:** Not applicable.

**Informed Consent Statement:** Not applicable.

**Data Availability Statement:** The data presented in this study are available in Supplementary Material.

**Acknowledgments:** We thank the editor G. Wang and the guest editors D. Liotta, G. Molli and A. Cipriani for the opportunity to share our research in the Special Issue "The Apennines: Tectonics, Sedimentation, and Magmatism from the Palaeozoic to the Present". Special thanks to the anonymous reviewers for their excellent contribution in restructuring and improving the manuscript.

**Conflicts of Interest:** The authors declare no conflict of interest.

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
