# Peer review of "Kinematics of Deformable Blocks: Application to the Opening of the Tyrrhenian Basin and the Formation of the Apennine Chain"

_geosciences, doi:10.3390/geosciences11040177_

Round 1

Reviewer 1 Report

Review of Turco et al. – Geosciences 2021

The manuscript of Turco et al. entitled “The opening of the Tyrrhenian basin and the Apennine chain formation in the kinematic context of Africa - Europe collision“ deals with the complex tectonic reconstructions of the Western-Central Mediterranean and evolution of the Apennines over the last 19 Ma. The topic and method used are very interesting and is of high interest for the understanding of the geodynamic evolution of this complex area. 

However, I have one major concern about the novelty of this work compared to what was already published by the same authors in 2013 (Turco et al., 2013) and in 2020 (Pierantoni et al. 2020). One principal aspect for a manuscript to be accepted for publication in Geosciences is: “Manuscripts should only report results that have not been submitted or published before, even in part”, thus my concern. The rotation parameters (Table 2) are the same as from Turco et al. (2013) and the kinematic maps and figure showing the slab configuration are almost the same as in Pierantoni et al. (2020), so what is new? This should be clarify/specify in the text.

The abstract, introduction and conclusion should be improved in that respect and should present what are the (new) scientific questions, novelty and main implications of this work. 

Other main comments:

(1) Kinematic constraints

My other main comment concerns the kinematic constraints used for the rotation model. The authors should show clearly which tectonic structures are used, at what time stage, for which sector. All figures showing the rotation grid and pole for each sector (11-12-13-14, as well as Figures 18-23-24) should be improved in that respect. Geographical (latitude-longitude frame, necessary to follow the text for example l. 340, l. 375, l. 396) and geological information should be added on all Figures. I would strongly suggest to actually show all these figures together in a “Method” or “Kinematic Parameters” section and then present the table with rotation parameters. 

We cannot assess the kinematic reconstructions if only the rotation parameters (Table 2) are given and not the structures that are reconstructed. The authors should provide (as supplementary file) the reconstructed features (as shapefiles and/or as .gpml files if they used GPlates) additionally to the table of rotation parameters. 

Concrete example – questions about reconstructions of sector 1:

 l. 620-622 and Figure 18: This part is not clear, the text refers to structures related to extension in the Tyrrhenian domain – whose orientation should be more clearly described, which basins exactly were formed during this stage and where, in which orientation? – between 19-12 Ma, while Figure 18 represents present-day topography and a pole grid at 5 Ma for the N. and Central Apennines. 

The orientation of extension within the Northern Apennines (sector 1) on Figure 19 does not seem to represent the orientation of the basins in this area (e.g. Radicondoli, Elsa, Siena and Radicofani basins ,etc.; e.g. Pascucci et al. 2007), which are striking rather NW-SE (visible in the topography, used apparently for calculating the rotation pole on Figure 11), thus with opposite direction of extension as shown here... It is confusing. Clarifications/explanation on the reconstructions – for all sectors - are needed.

(2) References to support the text

Within sections 4 and 5, references should be added throughout the text when describing specific deformation event and associated tectonic and sedimentary structures/record. 

All tectonic structures, mountains, basins etc. (especially key structures used in the rotation model) mentioned in the text should be clearly located on the figures. And vice versa, a figure should be referred to in the text when these structures are mentioned in the text. 

If not shown, at least the authors should add references so that the reader can find more information. Otherwise it’s very difficult to follow, and difficult to find the data supporting the rotation model and its interpretation.

For example (not exhaustive) “the 41st parallel line” (l. 133), “Ancona-Anzio line” (l. 249), Taormina Line (l. 774-780), etc.  Please check especially entire sections 4 and 5. 

Similarly in the ‘Introduction’ (l. 60-70), the authors only refer to their own previous works (except for Dewey et al. 1989), but many other kinematic reconstructions (based on kinematic constraints) have been proposed over the years for the Western Mediterranean area - Adriatic Plate (e.g. Romagny et al. 2020; Rosenbaum et al. 2002 Rosenbaum and Lister 2014; Van Hinsbergen et al. 2014 and 2020; Le Breton et al. 2017; some reconstructions are incorporated into global plate model: Hosseinpoor et al. 2016; Mueller et al. 2019). They should be mentioned and it should clearly state how this work differs , what it brings new

(3) Structure of the paper

Section 3 – Method: This section presents the general approach used for the kinematic reconstructions. The authors need to show more concretely what are the kinematic constraints used here, so basically what they show in their section 4 - “Results” (and this part should be improved, see above comment).

Section 4 “Results”: This section does not show the results but the kinematic constraints/assumptions of the rotation model. It could be merged with section 3 or renamed for example “Kinematic constraints and rotation model”.

A section “Results” should show the new kinematic reconstructions/maps. There, the figures should come in chronological order (5 Ma should come before 3 Ma). I would strongly recommend to put all reconstructions figures together (Figures 16, 17, 19, 20, 22, 25) with a common legend and list of abbreviations. We should not have to go to the last figure (who comes 7 pages later) to see the legend of the first map.

The section 5.2 “Slab-retreat Process” is a very interesting section as it discusses the implications of the rotation model in terms of geodynamic processes and how slab processes influence the surface, but is too short in its present version. The following section 5.3 actually discusses slab processes, so it’s redundant. I suggest to re-structure, make a short description of the main tectonic phases when presenting the kinematic maps in a section “Results” and then discuss the implications of the new rotation model in terms of tectonic and geodynamic (slab dynamics/processes) evolution of the area in the ‘Discussion’ section.

Specific comments:

l. 66 “the software”: which one? GPlates is mentioned but was this software used? If yes, I suggest the authors to provide the features (.gpml) and rotation files (.rot) as supplementary file.

l.68 and l. 846: “the global rotation model”: which one? 

l. 86 – l. 92 – Caption of Figures 2 and 3: caption for black and red triangles is missing

l. 130: migration of the “trench” (not subducting plate). The subducting plate is sinking/rolling back into the mantle, triggering a southward migration of the trench but the subducting plate is not migrating southward.

l. 138 – Figure 5: legend and reference for the rainbow background map is missing (crustal thickness?)

l. 148 - Figure 6: Reference for the magnetic anomaly map and Lat/ Lon values are missing

l. 164 – Figure 7: “modified after [32]” what are the modifications?

l. 187: reference to Greiner 1999 (Eulers rotations in plate tectonic reconstructions; Computers and Geosciences) is relevant here

Figures 11-12: What about the Corsica Basin? Why is it not included in the rotation grid of either the Northern Sector (Figure 11) or Central Sector (Figure 12)

l. 475: “finite stage pole”, finite or stage? I think the authors mean finite here.

l. 478: “absolute values” meaning “finite pole”?

l. 479: “using the balancing technique” and a pole of rotation (specify which one; which sector).

Figure 15: Legend and reference of the map (crustal thickness?) are missing. This figure could be completed to show how the rotation parameters are calculated from this balancing cross-section, by showing the location of the rotation pole on the map and the restored amount of motion between the two points (rather than the balanced area in km2).

l. 492-516 - Section 5.1.1.: This comes a bit out of topic since the paper concerns the opening of the Tyrrhenian Basin.

lines 504-505: juxtaposition of slab: reference? can we be so sure, no one can really differentiate parts and origin of slabs in tomographic models?

Line 515: “in agreement with” or based on?

l. 524: why does the rifting process “has to be” a few million years older?

l. 547-548: This was also proposed by Rosenbaum and Lister (2004).

l. 551: Please be consistent in the numbering or labelling of the different sectors in text and figures (1, 2, 3, 4, 5 or A, B, C, D ?)

l. 566-567: “rollback” could be mentioned here.

l. 591-592: reference for these marine successions?

l. 605-606 – Figure 17 caption: “Light blue lines are the Jurassic COB (Continent-Ocean Boundary)”: where? we only see transfer faults through the continental Adriatic plate in blue and the present-day coastlines. 

l. 626: which basins?

l. 684-685: This sentence is not clear (“the Calabrian arc rotates around its pole channeling into the Lagonegrese corridor”?)

l. 690 – Figure 21 caption: The number on this figure seems to indicate rather the 5 kinematic sectors, not the slab segments.

l. 781-785: “two rifts”: which rifts? 

l. 821: Conclusion is section 6. (not 5.)

The conclusion should summarise the main implications of these (new?) kinematic reconstructions 

English/Formulation:

Pronouns (it, this) and articles (the) are sometimes missing in sentences.

l. 82 (check throughout the text): “Adria Plate” should be “Adriatic Plate” (or use only “Adria”).

l.104: “actual” should be removed 

l. 112: Apennine “cover” should be changed to “chain”

l. 671: Ionian or Calabrian slab (not Tyrrhenian Slab)

l. 679, 775 (please check throughout text): Ionian not Ionic

“Left” – “Right” Transform should be changed to “Left-lateral (or sinistral)” and “Right-lateral (or dextral)”, for example in section 5.3.

Reviewer 2 Report

Dear Editor, 

Dear Authors,

I have reviewed the manuscript “The Opening of the Tyrrhenian Basin and the Apennine Chain  Formation in the Kinematic Context of Africa - Europe Collision” by Turco et al. 

It deals with the kinematic evolution of the western mediteranean, an area which experienced the Alpine collision and subsequent extension, in the frame of limited space, and Africa-Europe convergence. This makes the argument update and suitable to be of interest for a large international audience. 

The manuscript is divided in three parts: the first, is about the presentation of the data that will be used for the kinematic reconstruction; the second part is about the methodology; finally, the third part is about the results.

The introductory part is complete, although several oversights are still present. Figures need to be implemented. 

In the second part, the concept of micro-plates should be better disclosed in order to facilitate the reading and the focus of the text. Furthermore, several geographic names are cited in the text but not reported in the figures.

Conclusions are in agreement with the data and the methodological approach. The manuscript however needs from moderate to major revisions. In the attached file, my comments and punctual indications.

Hoping it helps, 

Best regards

Round 2

Reviewer 1 Report

The new version of the manuscript of E. Turco et al., now entitled “Kinematics of Deformable Blocks: Application to the Opening of the Tyrrhenian Basin and the Formation of the Apennines Chain” is significantly improved. The relation between the kinematics of the different sectors, the main structures observed along the Apennines and the dynamics of the subduction (slab tear/STEP fault) discussed here is very interesting. The authors provide an excellent overview of the evolution of the Apennines-Tyrrhenian Basin over the last 19 Ma. They have replied to most of my previous comments and made significant changes that improved the structure, presentation and discussion of the methods and kinematic reconstructions. The novelty, compared to Turco et al. (2013) and Pierantoni et al. (2020) lies on the detailed description of the method and the rotation model. In that regard, a few more points remain to be addressed (some important quantitative information/parameters are missing and uncertainties should be discussed, see main comment below), and thus require additional revision of the manuscript. I have also listed below minor comments regarding the figures, references and formulation/editing, which should be taken into account.

Main comment : Kinematic / rotation parameters: Section 3.3. and 3.4.6 (determination of angle of rotation + Table 1)

The approach is now well described, the grids to determine the pole for each sector are clearly presented. But, regarding the angle of rotation/amount of deformation, only one example for one sector is given (cross-section through Corsica-Umbria Marche). What about the other sectors and the deformation between each sector? These are crucial quantitative information on which the kinematic/rotation model lies that are currently missing: how much motion/deformation for each sector and between each sector were estimated and used in the kinematic reconstruction between the different key stages (19-12 Ma; 12-7 Ma; 7-3 Ma; 3-0 Ma)? It is not clear for example, how the complex shape of sector 3 (Figures 25-26-27) was exactly obtained/constrained through time.

Figures 11-12 could be removed as they can both be found in the references quoted. However, and instead of Fig. 11-12, a table synthetizing the key tectonic events (or a figure similar to the previous figure 9 in the first version of the manuscript), the key ages/timing used in the maps and rotation parameters (19, 12 , 7, 3 Ma) and providing quantitative information on the amount of motion/deformation for each sector should be presented here (section 3.3).

Regarding the presented profile (cross-section through Corsica-Umbria Marche, Fig. 20): Paragraph 492-503 and Figure 20 should be moved in section 3.3, where the authors describe how they estimate the angle of finite rotation based on the amount of extension obtained from crustal balancing. The reference for the present-day thickness should be indicated (I presume Grad et al. 2009, as on Figure 5 ?) and the uncertainty on the Moho depth – and thus on the resulting aerial balancing/amount of deformation along the profile - should be clearly stated (Grad et al. 2009 has c. 2 to 4 km uncertainties for this area). The assumption on the initial thickness of 40 km for the crust beneath Corsica should be also shortly explained/justified. The Liguro-Provencal Basin was already opened at 19 Ma, thus the crust beneath Corsica was most likely already thinned and no longer of ‘orogenic’ thickness of 40 km, why not using an initial thickness of 30 km? Today, the crust in the hinterland of the Apennines varies from 22-32 km (Figure 5). How much would that change the amount of displacement and angle of rotation?

Table 1: I presume that the lines of the table correspond to the lines delimitating each block, one on the Tyrrhenian side (named "Northern – Central – Southern – Sicilian sectors" and "Calabrian Arc") and one on the Apennines/Ionian side (named “Front”) as shown on Figure 19, is that correct? Please precise in the caption and/or name. However, the part of the table referring to “Lazio-Abruzzi” is not clear. Please clarify (to which line/sector it corresponds?). Also there, the reference ID 717 is used, please specify (Africa?).

Deformation between the blocks/sectors: did the authors calculated stage pole grid as shown on  Figure 15b and Figure 17 for each time interval? Why show here 5 Ma – present? Where are the morpho-structures on Figure 17a that should be compared to the grid?

As the authors wrote l. 919-920, yes this kinematic model would be very useful for future research, but all the quantitative information for the rotation model, or a link to a database/data repository, needs to be provided to be reproduced/used in the future. 

In the next sentence (l. 922), the authors mentioned an animation, where can it be found? Please provide the link. I have checked online the PCME of A. Schettino but it shows images of global reconstructions from the model of Schettino and Scotese (2004), not this detailed reconstruction of the Tyrrhenian – Apennines area.

Specific comments

Length / Number of figures – Sections 4 & 5:

The paper is very long and includes a lot of figures (28). Additionally to my comment on Figures 11 and 12 written above, I think that Figure 28 is not necessary. The different arcs of the northern Apennines are visible already on Figure 4, thus Figure 28 could be removed and Figure 4 quoted here (l. 867).

One possibility also to reduce the number of figures and length would be to have Figures 21 – 22 – 24 – 25 – 26 – 27 merged together into one figure (on one or two pages) with only one caption. It would avoid repeating the same caption and legend 6 times. Plus, it would provide a nice overview figure with the entire kinematic evolution of the area since 18 Ma. It would also allow to have as final figure of the paper, Figure 23 with the proposed slab geometry, which corresponds to the discussion in terms of STEP faults.

Sections 4 – 5: Either the description of the kinematics should be clearly separated from the implications into the sections Results and Discussion, respectively, which would avoid some redundancy between the two sections on the STEP fault evolution. Or, if kept as it is, section 4 “Results” should be renamed (e.g. “Kinematic reconstructions and geodynamic implications”), as this section presents not only results (the kinematic reconstructions) but also their implications / suggestions in terms of geodynamic evolution and slab tearing/STEP fault evolution.

References:

l. 60-61: the recent Mediterranean reconstruction of Van Hinsbergen et al. (2020; Gondwana Research, 81) and global compilation of Müller et al. (2019; Tectonics, 38), which also includes continental deformation along major plate boundaries, should be quoted here.

l. 399: Reference?

l. 800-804: Reference(s)?

Figures and captions:

Figure 4: add Northern Apennines

Figure 5: remove “and lithospheric”, this map shows only crustal thickness. Add caption for violet line (plate boundary).

Figure 17a: Gran Sasso front, Ancona-Anzio line and Laga basin mentioned in the text (l. 394) should be located on the figure

Figure 22: Gulf of Squillace mentioned l. 630-631 should be located on the figure

Figure 25: caption for V and CB should be added

Figures 21 – 22 – 24 – 25 – 26 – 27: rephrase “see this figure for the legend” (e.g. "see upper right legend"); remove “the Jurassic COB” for the light blue lines (light blue lines represent only continental transfer zones on these maps).

Formulation/Editing:

Abstract l. 18-19 -> the sentence “The deformation of tectonics elements…” should come later at l. 24 (before last sentence), and “velocities of points that compose these blocks” should be changed to “velocities of lines that delimitates these blocks (on both Tyrrhenian and Apennines sides)” (no? if points, then please provide more information in the method section and table 1).

The authors are sometimes too affirmative when presenting the implications of their models in terms of STEP fault evolution (sections 4 and 5). More suggestive formulation like “Our model proposes/suggests/implies…” should be used. Examples (should be checked throughout these two sections) l. 590-592; l. 604-606; l. 804-806; sections 5.3 and 5.4.

Caption Figure 23: “Proposed Ligurian-Ionian slab geometric reconstruction”.

Title sections 5.2 - 5.3 - 5.4: rephrase “The role of”, e.g. “STEP fault evolution within the…”, (or precise role on what?)

l. 211: palinspastic

l. 260: ad -> and

l. 299: by taking (remove “the”)

l. 827-828: this sentence should be rephrased, e.g. “The method described in this work is based on the laws of plate kinematics and takes into account deformation of blocks through time.”

l. 830: shed new light on (not ‘to’) the geodynamic processes

Reviewer 2 Report

Dear Editor,

Dear Authors,

I have newly revised the manuscript “Kinematics of Deformable Blocks: Application to the Opening of the Tyrrhenian Basin and the Formation of the Apennine Chain” submitted by Turco et al.

The authors have accepted my previous hints and the new text is improved and ready for publication, although a reference is missing.

A few minor oversights and typos can be solved during the proofs preparation stage. 

My suggestions are included in the attached file.

Kind regards

Author Response

We have accepted all corrections made by the reviewer